# Variational Bayesian Flow Network for Graph Generation

Yida Xiong [1] [*]   Jiameng Chen [1] [*]   Xiuwen Gong [1]   Jia Wu [2]   Shirui Pan [3]   Wenbin Hu [1] [4]

## Abstract

Graph generation aims to sample discrete node and edge attributes while satisfying coupled structural constraints. Diffusion models for graphs often adopt largely factorized forward-noising, and many flow-matching methods start from factorized reference noise and coordinate-wise interpolation, so node–edge coupling is not encoded by the generative geometry and must be recovered implicitly by the core network, which can be brittle after discrete decoding. Bayesian Flow Networks (BFNs) evolve distribution parameters and naturally support discrete generation. But classical BFNs typically rely on factorized beliefs and independent channels, which limit geometric evidence fusion. We propose *Variational Bayesian Flow Network* (VBFN), which performs a variational lifting to a tractable joint Gaussian variational belief family governed by structured precisions. Each Bayesian update reduces to solving a symmetric positive definite linear system, enabling coupled node and edge updates within a single fusion step. We construct sample-agnostic sparse precisions from a representation-induced dependency graph, thereby avoiding label leakage while enforcing node–edge consistency. On synthetic and molecular graph datasets, VBFN improves fidelity and diversity, and surpasses baseline methods. The reproducible code is available at https://github.com/Cello2195/VBFN.

## 1. Introduction

Graph generation is a fundamental problem with broad impact in network science (Xu et al., 2019; Su et al., 2022),

*Equal contribution [1]School of Computer Science, Wuhan University, Wuhan, China [2]Department of Computing, Macquarie University, Sydney, Australia [3]Griffith University, Gold Coast, Australia [4]Wuhan University Shenzhen Research Institute, Shenzhen, China. Correspondence to: Wenbin Hu <hwb@whu.edu.cn>.

*Proceedings of the $43^{rd}$ International Conference on Machine Learning*, Seoul, South Korea. PMLR 306, 2026. Copyright 2026 by the author(s).

drug discovery (Li et al., 2025), and computational biology (Chen et al., 2025). Diffusion models (Jo et al., 2024; Boget, 2025) and flow matching methods (Eijkelboom et al., 2024; Qin et al., 2025) have achieved strong performance on graph-structured data (Liu et al., 2023; Cao et al., 2024; Xiong et al., 2025; Li et al., 2026). However, graph generation remains challenging because graphs are discrete and their validity depends on coupled relations between node variables and edge variables.

Many graph diffusion and flow matching methods learn in a continuous surrogate space by regressing embeddings or logits. They recover discrete node and edge attributes during sampling using an $\arg\max$ rule or a projection operator. Discrete-state diffusion avoids this design by defining forward transitions directly on categorical spaces (Austin et al., 2021). When training relies on a continuous regression loss, but sampling is governed by discrete decisions, the surrogate objective will prioritize within-class numerical fidelity and encourage averaging behaviors that increase the chance of boundary errors after discretization (Bartlett et al., 2006). In relational settings, a few boundary errors can violate global consistency constraints and distort structural statistics.

Another limitation is the geometry induced by the noising process. Many diffusion models start from factorized isotropic noise and use Gaussian perturbations with independent coordinates (Ho et al., 2020; Song et al., 2021). This independence geometry fails to provide an explicit probabilistic mechanism for joint evidence fusion across node and edge variables, so geometric coupling must be reconstructed by the core network or enforced by external penalties, which increases the burden on model capacity.

Bayesian Flow Networks (BFNs) (Graves et al., 2023) provide a complementary viewpoint by evolving beliefs in the space of distribution parameters and matching sender and receiver message distributions through Bayesian update. This makes discrete generation natural because beliefs parameterize distributions whose integrated mass gives class probabilities, so sampling does not rely on an external rounding step (Graves et al., 2023; Xiong et al., 2026). BFNs further support flexible channel designs and apply to continuous, discrete, and discretized modalities (Song et al., 2024; Qu et al., 2024). Nevertheless, classical BFNs typically assume a factorized belief family with an independent noise channel.

As a result, Bayesian update is performed element-wise and geometric dependencies must be learned implicitly by the core network or enforced by external regularization.

To address this limitation, we propose *Variational Bayesian Flow Network* (VBFN), which performs a variational lifting from factorized Gaussian beliefs to a tractable joint Gaussian variational belief family governed by structured precisions. VBFN introduces correlation directly into the graph generation and fusion mechanism, and it turns Bayesian update from an element-wise rule into a coupled inference problem. At each time $t$, the posterior belief is obtained by solving a single symmetric positive definite (SPD) linear system, which enables coherent joint updates of node and edge variables within one computation. VBFN preserves the message–matching principle of classical BFNs and it keeps the continuous-time training objective analytically tractable while replacing independence geometry with a relational one. Particularly, we parameterize the sender channel with a tied covariance structure across time, which yields a closed-form structured variational objective while retaining a coupled update geometry. The coupling is defined by a representation-induced dependency graph and it never queries the unknown label adjacency, which avoids label leakage while enforcing basic node and edge consistency. We summarize our contributions as follows.

- We lift the factorized BFN belief family to a tractable joint Gaussian variational belief family governed by structured precision operators, enabling geometric belief evolution for graph generation.

- We derive a closed-form joint Bayesian update in operator form, where posterior inference reduces to solving an SPD linear system that couples node and edge variables within each update.

- We keep the message–matching principle of classical BFNs and derive a structured variational objective under a tied channel covariance, in which the flow distribution is induced by coupled update dynamics.

## 2. Related Work

**Graph generative models.** Autoregressive graph generators produce nodes and edges sequentially to maintain validity (You et al., 2018; Liao et al., 2019; Dai et al., 2020; Goyal et al., 2020), but their inference is costly. One-shot GAN and VAE variants improve parallelism (Goodfellow et al., 2014; Kingma & Welling, 2013) yet often struggle with geometric dependencies under simplified assumptions (Martinkus et al., 2022; Prykhodko et al., 2019; Li et al., 2022; Huang et al., 2022). Recent work is dominated by diffusion and flow-based paradigms.

**Diffusion and flow matching.** Diffusion models have become a leading approach for graph generation in both score-based and discrete formulations (Jo et al., 2022; Vignac et al., 2023; Xu et al., 2024; Lee et al., 2023). Flow matching methods provide an alternative continuous-time training principle (Eijkelboom et al., 2024; Qin et al., 2025). Many methods employ factorized noise that perturbs node and edge attributes independently, which does not encode intrinsic coupling and leaves structural consistency to be recovered by the core network (Jo et al., 2024).

**Bayesian flow networks.** BFNs evolve distribution parameters and match sender and receiver message distributions through Bayesian update (Graves et al., 2023; Qu et al., 2024). Classical BFNs typically rely on factorized coding geometry for tractable updates, which limits their ability to represent geometric structure (Xiong et al., 2026).

## 3. Background

### 3.1. Problem Definition

We study generative modeling of graphs $G = (X, A)$ with $N$ nodes drawn from an unknown distribution $p_{\text{data}}(G)$, where $X \in \mathbb{R}^{N \times d_x}$ denotes node attributes and $A \in \mathbb{R}^{N \times N \times d_a}$ denotes edge attributes. This formulation covers general graphs whose attributes may be continuous or categorical. We represent them in a unified continuous parameterization $\boldsymbol{z} \in \mathbb{R}^D$ where $D = N d_x + N^2 d_a$, and keep task-specific parameterization details in Appendix B.

### 3.2. Bayesian Flow Networks

BFNs define a generative process as a continuous-time *Bayesian inference* process on a latent signal representation. Rather than defining a forward-noising dynamics on samples like diffusion models, BFNs maintain a time-indexed belief $q_t(\boldsymbol{z})$ and refine it by repeatedly fusing Gaussian messages through Bayesian update. The essential point is that the stochastic process evolves in the *space of belief parameters* rather than directly evolving a noisy sample.

**Belief parameter.** In Gaussian BFNs, let $t \in [0, 1]$ be the process time, and the belief parameter over the unknown signal $\boldsymbol{z} \in \mathbb{R}^D$ is chosen to be an independent Gaussian

$$q_t(\boldsymbol{z}) = \mathcal{N}\big(\boldsymbol{z}; \boldsymbol{\mu}_t, \rho_t^{-1} \boldsymbol{I}\big), \tag{1}$$

where $\boldsymbol{\theta}_t \triangleq \{\boldsymbol{\mu}_t, \rho_t\} \in \mathbb{R}^D$ collects the mean $\boldsymbol{\mu}_t \in \mathbb{R}^D$ and a scalar precision $\rho_t > 0$. And variance $\sigma_t^2 \triangleq \rho_t^{-1}$. Equivalently, the belief precision is $\boldsymbol{\Sigma}_t^{-1} = \sigma_t^{-2} \boldsymbol{I}$. This factorized coding geometry is the key restriction that enables closed-form Bayesian update in classical BFNs.

**Sender and receiver distributions.** BFNs introduce a message variable $\boldsymbol{y} \in \mathbb{R}^D$. The sender reveals information

about the unknown target $z$ by transmitting a noisy message distribution:

$$p_{\mathrm{S}}(\boldsymbol{y} \mid \boldsymbol{z}; t) = \mathcal{N}\Big(\boldsymbol{y}; \boldsymbol{z}, \alpha(t)^{-1}\boldsymbol{I}\Big), \qquad (2)$$

where $\alpha(t) > 0$ is an accuracy rate. The *receiver* forms a matched Gaussian distribution centered at the model prediction $\widehat{\boldsymbol{z}}_\phi(\boldsymbol{\theta}_t, t)$:

$$p_{\mathrm{R}}(\boldsymbol{y} \mid \boldsymbol{\theta}_t; t) = \mathcal{N}\Big(\boldsymbol{y}; \widehat{\boldsymbol{z}}_\phi(\boldsymbol{\theta}_t, t), \alpha(t)^{-1}\boldsymbol{I}\Big). \qquad (3)$$

Intuitively, $p_{\mathrm{S}}$ describes how the sender transmits a measurement of $z$ at accuracy $\alpha(t)$, while $p_{\mathrm{R}}$ describes the distribution of messages implied by the model's current belief.

**Bayesian update and flow distribution.** Given a noisy message $\boldsymbol{y} \sim p_{\mathrm{S}}(\cdot \mid \boldsymbol{z}; t)$, BFNs update the belief parameter by Bayesian update:

$$q_t^+(\boldsymbol{z}) \; \propto \; q_t(\boldsymbol{z})\, p_{\mathrm{S}}(\boldsymbol{y} \mid \boldsymbol{z}; t). \qquad (4)$$

Because both factors are Gaussian in $z$, the posterior remains Gaussian and the Bayesian update admits a closed form. Specifically, $\boldsymbol{\theta}_t^+ \triangleq \{\boldsymbol{\mu}_t^+, \rho_t^+\}$ is updated via

$$\begin{aligned} \rho_t^+ &= \rho_t + \alpha(t), \\ \boldsymbol{\mu}_t^+ &= \frac{\rho_t\, \boldsymbol{\mu}_t + \alpha(t)\, \boldsymbol{y}}{\rho_t + \alpha(t)}. \end{aligned} \qquad (5)$$

Starting from a prior $q_0$, repeatedly sampling $\boldsymbol{y}$ from the sender and applying Eq. (5) defines a stochastic process of belief parameters $\boldsymbol{\theta}_t$. The distribution of $\boldsymbol{\theta}_t$ induced by a given target $z$ is referred to as the *flow distribution*:

$$\boldsymbol{\theta}_t \sim p_F(\cdot \mid \boldsymbol{z}; t). \qquad (6)$$

**Training objective.** BFNs train the predictor by matching sender and receiver message distributions along the flow:

$$\begin{aligned} \mathcal{L}_\infty(\phi) =& \mathbb{E}_{\boldsymbol{z} \sim p_{\mathrm{data}}} \int_0^1 \mathbb{E}_{\boldsymbol{\theta}_t \sim p_F(\cdot \mid \boldsymbol{z}; t)} \\ & \Big[ D_{\mathrm{KL}}\big(p_{\mathrm{S}}(\cdot \mid \boldsymbol{z}; t) \,\big\|\, p_{\mathrm{R}}(\cdot \mid \boldsymbol{\theta}_t; t)\big) \Big]\, d\beta(t). \end{aligned} \qquad (7)$$

Because Eqs. (2) and (3) tie the covariance, the Gaussian KL divergence simplifies to a weighted squared error between $z$ and $\widehat{\boldsymbol{z}}_\phi(\boldsymbol{\theta}_t, t)$, yielding the familiar $\mathcal{L}_\infty$ objective.

**Limitation for graphs.** Classical BFNs are analytically convenient since both the belief uncertainty and the sender channel are independent. As a result, the Bayesian update decomposes into independent reweighting across indices. For graphs, this independent geometry fails to encode instantaneous dependencies between $X$ and $A$. VBFN keeps the message–matching objective but lifts this geometry to joint precisions so that inference becomes intrinsically coupled.

## 4. Methodology

### 4.1. Variational Lifting of Factorized Geometry

We introduce a joint Gaussian variational belief family with structured precision, which keeps Bayesian update tractable while turning element-wise fusion into coupled inference for node and edge variables. For geometric graph data, the independence geometry intrinsic to classical BFNs fails to model the dependencies between nodes and edges, since node and edge degrees of freedom are inherently coupled. VBFN departs from this independence geometry and lifts the belief family from independent Gaussians to a variational family of joint Gaussians whose uncertainty is governed by a precision operator.

Concretely, VBFN lifts the factorized family to a tractable joint Gaussian belief family over the graph signal $z$:

$$q_t(\boldsymbol{z}) = \mathcal{N}\big(\boldsymbol{z}; \boldsymbol{\theta}_t, \boldsymbol{P}_t^{-1}\big), \qquad (8)$$

where $\boldsymbol{P}_t = \boldsymbol{\Sigma}_t^{-1} \succ \boldsymbol{0}$ is a time-dependent joint precision, which governs how evidence about one component of the graph signal influences all others during Bayesian update. This restriction preserves closed-form updating while introducing geometric coupling through $\boldsymbol{P}_t$.

We parameterize information revelation by an accuracy rate $\alpha(t) > 0$ and define the cumulative schedule

$$\beta(t) \triangleq \int_0^t \alpha(s)\, ds, \qquad (9)$$

so that $\beta(0) = 0$. A predictor $\widehat{\boldsymbol{z}}_\phi(\boldsymbol{\theta}_t, t)$ produces a denoised estimate of $z$ from the current state. The remaining subsections specify how we instantiate structural operators and how they induce a coupled flow distribution $p_F(\boldsymbol{\theta}_t \mid \boldsymbol{z}; t)$ through Bayesian update. The framework of VBFN is illustrated in Figure 1.

### 4.2. Intrinsic Structural Priors

**The necessity of structural geometry.** A joint belief parameter becomes meaningful only if the geometry that defines it is itself structural. VBFN therefore introduces a structured precision operator $\boldsymbol{\Omega}_{\mathrm{prior}} \succ \boldsymbol{0}$ and equips the latent graph signal $z$ with a Gaussian Markov random field (GMRF) prior (Rue & Held, 2005)

$$p_{\mathrm{prior}}(\boldsymbol{z}) = \mathcal{N}\big(\boldsymbol{z}; \boldsymbol{\theta}_0, \boldsymbol{\Omega}_{\mathrm{prior}}^{-1}\big). \qquad (10)$$

In graph generation, $\boldsymbol{\Omega}_{\mathrm{prior}}$ must not depend on the unknown sample adjacency $A$ to avoid label leakage. We instead construct $\boldsymbol{\Omega}_{\mathrm{prior}}$ from a fixed dependency structure that is induced by the representation of $(X, A)$ itself. This structure captures how node and edge variables must interact under basic consistency relations, while remaining agnostic to which edges are present in any particular sample.

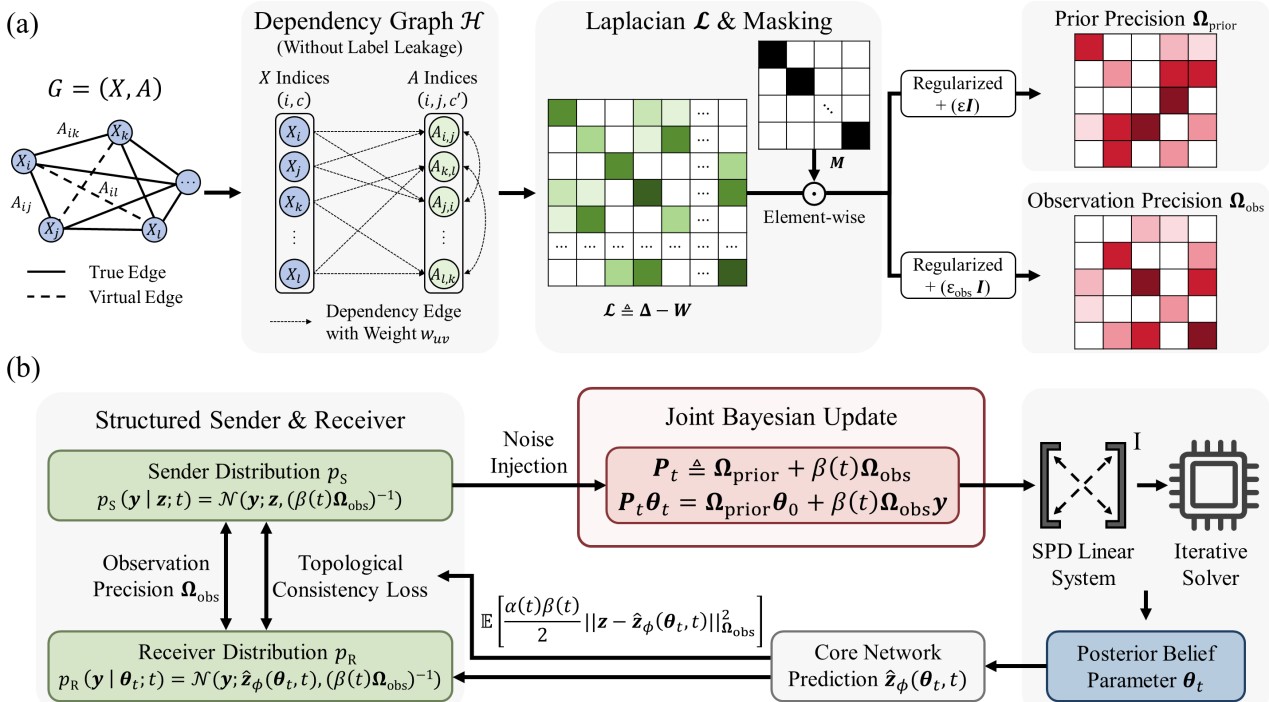

**Figure 1.** Illustration of VBFN framework. **(a)**: From $G = (X, A)$, construct a no-leakage dependency graph $\mathcal{H}$, assign edge weights $\{w_{uv}\}$, and form the masked weighted Laplacian $\mathcal{L}$. Then instantiate $\Omega_{\text{prior}}$ and obtain $\Omega_{\text{obs}}$. **(b)**: With fixed message precision $\Omega_{\text{obs}}$, sender–receiver message–matching defines the training loss, while inference uses a coupled Bayesian update solved by iterative solvers.

**A representation-level dependency graph.** Let $\mathcal{I}_X = \{(i, c) : i \in [N], c \in [d_x]\}$ index entries of $X$, and $\mathcal{I}_A = \{(i, j, c) : i, j \in [N], c \in [d_a]\}$ index entries of $A$. Define a dependency graph $\mathcal{H} = (\mathcal{V}_{\mathcal{H}}, \mathcal{E}_{\mathcal{H}})$ with vertices $\mathcal{V}_{\mathcal{H}} = \mathcal{I}_X \cup \mathcal{I}_A$, and edges capturing intrinsic representation couplings:

$$\mathcal{E}_{\text{inc}} = \Big\{ \big((i, c), (i, j, c')\big), \ \big((j, c), (i, j, c')\big) \\ : (i, c) \in \mathcal{I}_X, \ (i, j, c') \in \mathcal{I}_A \Big\}. \quad (11)$$

We also include symmetry couplings when the chosen parameterization enforces $A_{ij} = A_{ji}$ by adding

$$\mathcal{E}_{\text{sym}} = \Big\{ \big((i, j, c), (j, i, c)\big) : (i, j, c) \in \mathcal{I}_A, \ i < j \Big\}, \quad (12)$$

and setting $\mathcal{E}_{\mathcal{H}} = \mathcal{E}_{\text{inc}} \cup \mathcal{E}_{\text{sym}}$; otherwise $\mathcal{E}_{\text{sym}} = \emptyset$.

Crucially, $\mathcal{H}$ depends only on the tensor structure of $(X, A)$ and on invariances of the representation, and it consequently never queries the sample-specific adjacency values.

We equip the dependency graph $\mathcal{H} = (\mathcal{V}_{\mathcal{H}}, \mathcal{E}_{\mathcal{H}})$ with non-negative edge weights $\{w_{uv}\}_{(u,v)\in\mathcal{E}_{\mathcal{H}}}$, where $w_{uv} = w_{vu}$ for an undirected $\mathcal{H}$. Unless stated otherwise, we set $w_{uv} = \lambda_X$ for $(u, v) \in \mathcal{E}_{\text{inc}}$ and $w_{uv} = \lambda_A$ for $(u, v) \in \mathcal{E}_{\text{sym}}$, with $\lambda_X, \lambda_A \geq 0$. Then define the weighted adjacency matrix

$W \in \mathbb{R}^{D \times D}$ as well as the degree matrix $\Delta \in \mathbb{R}^{D \times D}$ by

$$W_{uv} = \begin{cases} w_{uv}, & \text{if } (u, v) \in \mathcal{E}_{\mathcal{H}}, \\ 0, & \text{otherwise}, \end{cases}$$
$$\Delta_{uu} = \sum_{v \in \mathcal{V}_{\mathcal{H}}} W_{uv}. \quad (13)$$

Intuitively, $\Delta_{uu}$ aggregates the total coupling strength incident to index $u$ in $\mathcal{H}$.

**Weighted combinatorial Laplacian.** The weighted combinatorial Laplacian of $\mathcal{H}$ is the linear operator

$$\mathcal{L} \triangleq \Delta - W \in \mathbb{R}^{D \times D}, \quad (14)$$

which acts on any signal $s \in \mathbb{R}^D$ indexed by $\mathcal{V}_{\mathcal{H}}$ via $(\mathcal{L}s)_u = \sum_v W_{uv}(s_u - s_v)$.

With the diagonal mask operator $M$, we instantiate the prior precision as

$$\Omega_{\text{prior}} = M \mathcal{L} M + \varepsilon I, \quad (15)$$

where $(\lambda_X, \lambda_A)$ are encoded in $W$ and $\varepsilon > 0$. Placing $\varepsilon I$ outside the masking guarantees $\Omega_{\text{prior}} \succ 0$ even when $M$ zeroes padded entries, ensuring that posterior inference remains mathematically solvable.

The Laplacian also defines a nonnegative quadratic functional that measures disagreement along the couplings in $\mathcal{H}$.

For any $s \in \mathbb{R}^D$, define the Dirichlet energy

$$\mathcal{J}(s) \triangleq s^\top \mathcal{L} s = \frac{1}{2} \sum_{(u,v) \in \mathcal{E}_\mathcal{H}} w_{uv}(s_u - s_v)^2. \quad (16)$$

A Laplacian-based precision therefore does not enforce global averaging across all entries of $(X, A)$. It penalizes discrepancies only between pairs $(u, v)$ adjacent in $\mathcal{H}$, and the strengths $(\lambda_X, \lambda_A)$ in $\{w_{uv}\}$ control how strongly this structural bias influences the prior relative to the subsequent data-driven Bayesian update.

### 4.3. Joint Bayesian Update via a Structured Channel

A structural prior alone is insufficient: if the transmission channel injects independent noise, the Bayesian update remains effectively local.

**Structured sender and receiver.** VBFN therefore defines a structured sender whose noise is colored by an observation precision $\Omega_\mathrm{obs} \succ 0$ that shapes how uncertainty is injected into the transmitted messages. To avoid label leakage, we construct $\Omega_\mathrm{obs}$ from the same fixed dependency structure used for $\Omega_\mathrm{prior}$, either by tying it to $\Omega_\mathrm{prior}$ or by adopting a diagonalized variant for numerical stability. The concrete instantiations and configuration choices are in Appendix D.4.

By time $t$, the sender transmits a cumulative message $y$ via

$$p_\mathrm{S}(y \mid z; t) = \mathcal{N}\Big(y; z, \big(\beta(t)\Omega_\mathrm{obs}\big)^{-1}\Big). \quad (17)$$

The receiver constructs an analogous message distribution centered at its prediction:

$$p_\mathrm{R}(y \mid \boldsymbol{\theta}_t; t) = \mathcal{N}\Big(y; \widehat{z}_\phi(\boldsymbol{\theta}_t, t), \big(\beta(t)\Omega_\mathrm{obs}\big)^{-1}\Big). \quad (18)$$

**Bayesian update rule and coupled fusion.** Posterior inference is defined by the Bayesian product rule

$$p_\mathrm{post}(z \mid y; t) \propto p_\mathrm{prior}(z)\, p_\mathrm{S}(y \mid z; t). \quad (19)$$

Both factors are Gaussian in $z$, so their product is Gaussian. In canonical form, multiplying Gaussians adds their quadratic terms, which implies that the corresponding precision operators add. This is the point at which the update becomes intrinsically coupled.

The Bayesian product rule Eq. (19) implies that posterior inference remains within the Gaussian family. The following result states the closed-form posterior and reveals that, under structured precisions, the Bayesian update becomes a coupled linear solve.

**Theorem 4.1** (Joint Bayesian update with structured precisions). *Assume the structured prior Eq. (10) with $\Omega_\mathrm{prior} \succ$*

*0 and the structured sender Eq. (17) with $\Omega_\mathrm{obs} \succ 0$. Define the fused posterior precision at time $t$*

$$P_t \triangleq \Omega_\mathrm{prior} + \beta(t)\Omega_\mathrm{obs}. \quad (20)$$

*Then $P_t \succ 0$ and*

$$p_\mathrm{post}(z \mid y; t) = \mathcal{N}\big(z; \boldsymbol{\theta}_t, P_t^{-1}\big), \quad (21)$$

*where the posterior belief parameter $\boldsymbol{\theta}_t$ is the unique solution of the SPD linear system*

$$P_t\, \boldsymbol{\theta}_t = \Omega_\mathrm{prior}\boldsymbol{\theta}_0 + \beta(t)\Omega_\mathrm{obs}y. \quad (22)$$

Theorem 4.1 is the mathematical point where VBFN departs from factorized BFNs. When $\Omega_\mathrm{prior}$ and $\Omega_\mathrm{obs}$ are diagonal, Eq. (22) decomposes into independent scalar fusions. When they are structured, Eq. (22) performs a coupled equilibrium computation. Even if $P_t$ is sparse, its inverse is typically dense, so a local perturbation in $y$ propagates globally through $P_t^{-1}$ within a single Bayesian update. This is the operational meaning of co-evolution. Explicit inversion is unnecessary, and Sec. 4.5 shows how to solve Eq. (22) efficiently using iterative methods and matrix–vector products with $P_t$. The complete proof is provided in Appendix A.1.

The following Proposition 4.2 clarifies how VBFN generalizes classical Gaussian BFNs.

**Proposition 4.2** (Classical BFNs as a special case). *Suppose $\Omega_\mathrm{prior}$ and $\Omega_\mathrm{obs}$ are diagonal, namely*

$$\Omega_\mathrm{prior} = \mathrm{diag}(\omega_1^\mathrm{prior}, \ldots, \omega_D^\mathrm{prior}),$$
$$\Omega_\mathrm{obs} = \mathrm{diag}(\omega_1^\mathrm{obs}, \ldots, \omega_D^\mathrm{obs}). \quad (23)$$

*Then Eq. (22) decomposes across dimensions. For each $i \in [D]$, the posterior belief parameter satisfies the scalar fusion rule*

$$\theta_{t,i} = \frac{\omega_i^\mathrm{prior}\, \theta_{0,i} + \beta(t)\, \omega_i^\mathrm{obs}\, y_i}{\omega_i^\mathrm{prior} + \beta(t)\, \omega_i^\mathrm{obs}},$$
$$P_{t,i} = \omega_i^\mathrm{prior} + \beta(t)\, \omega_i^\mathrm{obs}, \quad (24)$$

*which coincides with the standard factorized Gaussian BFN update when $\omega_i^\mathrm{obs} \equiv 1$.*

**Proposition 4.3** (Posterior belief parameter as a coupled energy minimizer). *Under the assumptions of Theorem 4.1, the posterior belief parameter satisfies*

$$\boldsymbol{\theta}_t = \arg\min_{u \in \mathbb{R}^D} \left\{ \frac{1}{2}\|u - \boldsymbol{\theta}_0\|_{\Omega_\mathrm{prior}}^2 + \frac{\beta(t)}{2}\|y - u\|_{\Omega_\mathrm{obs}}^2 \right\},$$
$$\|v\|_\Omega^2 \triangleq v^\top \Omega v. \quad (25)$$

*If $\Omega_\mathrm{prior}$ is Laplacian-regularized as in Eq. (15), the first term in Eq. (25) contains the Dirichlet form Eq. (16) and penalizes structurally incoherent variation across representation-coupled variables, while the second term drives the belief parameter towards the data distribution.*

## 4.4. Training Objective

The variational objective matches sender and receiver messages in KL. Although the KL between general Gaussians contains trace and log-determinant terms, tying the channel covariance in Eqs. (17) and (18) cancels those terms and yields a tractable structured quadratic.

A technical point concerns the definiteness of the induced quadratic. Throughout, we require $\mathbf{\Omega}_{\text{obs}} \succ \mathbf{0}$ so that $\|\cdot\|_{\mathbf{\Omega}_{\text{obs}}}$ is a true norm. When $\mathbf{\Omega}_{\text{obs}}$ is instantiated from a Laplacian-based operator, we include a diagonal regularization term to remove the Laplacian null space and prevent spectral collapse. The concrete construction is in Appendix B.3.

**Proposition 4.4** (Structured KL under tied channel covariance). *For any fixed $t$ and state $\boldsymbol{\theta}_t$,*

$$
\begin{aligned}
&D_{\text{KL}}\Big( p_{\text{S}}(\cdot \mid \boldsymbol{z}; t) \,\Big\|\, p_{\text{R}}(\cdot \mid \boldsymbol{\theta}_t; t) \Big) \\
&= \frac{\beta(t)}{2}\|\boldsymbol{z} - \widehat{\boldsymbol{z}}_\phi(\boldsymbol{\theta}_t, t)\|_{\mathbf{\Omega}_{\text{obs}}}^2.
\end{aligned}
\tag{26}
$$

Substituting Eq. (26) into the flow objective Eq. (7) yields

$$
\begin{aligned}
\mathcal{L}_\infty(\phi) =\, &\mathbb{E}_{\boldsymbol{z} \sim p_{\text{data}}} \int_0^1 \mathbb{E}_{\boldsymbol{\theta}_t \sim p_F(\cdot \mid \boldsymbol{z}; t)} \\
&\left[ \frac{\alpha(t)\beta(t)}{2} \|\boldsymbol{z} - \widehat{\boldsymbol{z}}_\phi(\boldsymbol{\theta}_t, t)\|_{\mathbf{\Omega}_{\text{obs}}}^2 \right] dt.
\end{aligned}
\tag{27}
$$

In particular, the structural geometry affects learning through the weighted quadratic $\|\cdot\|_{\mathbf{\Omega}_{\text{obs}}}^2$, while the flow distribution $p_F(\cdot \mid \boldsymbol{z}; t)$ is generated by the coupled Bayesian update dynamics in Theorem 4.1.

To make the continuous-time limit explicit, consider a partition $0 = t_0 < \cdots < t_T = 1$, and define $\Delta\beta_k = \beta(t_k) - \beta(t_{k-1})$. At step $k$, the receiver fuses the next message with the previous belief parameter, and we work in canonical parameters. In canonical parameters, Theorem 4.1 implies the additive recursion

$$
\begin{aligned}
\boldsymbol{P}_k &= \boldsymbol{P}_{k-1} + \Delta\beta_k\, \mathbf{\Omega}_{\text{obs}}, \\
\boldsymbol{h}_k &= \boldsymbol{h}_{k-1} + \Delta\beta_k\, \mathbf{\Omega}_{\text{obs}}\boldsymbol{y}_k, \\
\boldsymbol{\theta}_k &= \boldsymbol{P}_k^{-1}\boldsymbol{h}_k,
\end{aligned}
\tag{28}
$$

where $\boldsymbol{P}_0 = \mathbf{\Omega}_{\text{prior}}$. The stepwise KL takes the form $\frac{\beta(t_{k-1})}{2}\|\boldsymbol{z} - \widehat{\boldsymbol{z}}_\phi(\boldsymbol{\theta}_{k-1}, t_{k-1})\|_{\mathbf{\Omega}_{\text{obs}}}^2$ under tied covariance $(\beta(t_{k-1})\mathbf{\Omega}_{\text{obs}})^{-1}$. Weighting each step by $\Delta\beta_k$ yields a Riemann approximation which converges in probability to Eq. (27) as $\max_k(t_k - t_{k-1}) \to 0$ and $\Delta\beta_k \approx \alpha(t_k)(t_k - t_{k-1})$. This yields the continuous-time objective without omitting the natural-parameter step that generates the flow.

Formally, VBFN lifts the variational family from factorized to joint Gaussians. We fix $\mathbf{\Omega}_{\text{obs}}$ to enforce a geometric prior, prioritizing the analytic tractability of the continuous-time limit over informative covariance.

## 4.5. Solving SPD Linear System of Eq. (22)

The computational core of VBFN is the joint Bayesian update in Eq. (22), which requires solving an SPD linear system with the fused precision $\boldsymbol{P}_t$. A dense factorization would cost $O(D^3)$ time and $O(D^2)$ memory, which is prohibitive since $D = Nd_x + N^2 d_a$ scales quadratically with $N$ for edge-rich representations. VBFN is designed so that $\mathbf{\Omega}_{\text{prior}}$ and $\mathbf{\Omega}_{\text{obs}}$ admit fast matrix–vector products (MVPs), enabling operator-form inference without materializing dense matrices.

For Laplacian-regularized constructions on $\mathcal{H}$, the operator $\mathcal{L}$ has $O(|\mathcal{E}_\mathcal{H}|)$ nonzeros. Under bounded-degree coupling templates such as incidence and symmetry, $|\mathcal{E}_\mathcal{H}|$ is linear in $D$. Consequently, an MVP $\boldsymbol{v} \mapsto \boldsymbol{P}_t \boldsymbol{v}$ costs $O(D)$, which is $O(N^2)$ for typical graph representations.

**Solver choice.** We provide two solvers for Eq. (22). For small or moderately sized graphs, we can apply a direct Cholesky (Golub & Van Loan, 2013) factorization to obtain an exact solution. For scalability, our default solver is Conjugate Gradient (CG) (Hestenes et al., 1952), which only requires MVPs and stores a constant number of vectors. A comparison between CG and Cholesky in accuracy is reported in Sec. 5.3 and Appendix C.2.

**Iteration complexity and conditioning.** After $K$ CG iterations, the per-update cost is $O(K \cdot \text{mv}(\boldsymbol{P}_t))$, hence $O(KD)$ under sparse operator realizations (Saad, 2003). The iteration count is governed by the condition number $\kappa(\boldsymbol{P}_t)$. To reach relative residual $\delta$, CG satisfies the standard rate

$$
K = O\Big( \sqrt{\kappa(\boldsymbol{P}_t)} \log(1/\delta) \Big).
\tag{29}
$$

The spectral floors in Eqs. (15) and (64) are therefore not merely for positive definiteness; they also stabilize $\kappa(\boldsymbol{P}_t)$. Indeed, for any $t$,

$$
\kappa(\boldsymbol{P}_t) \leq \frac{\lambda_{\max}(\mathbf{\Omega}_{\text{prior}}) + \beta(t)\lambda_{\max}(\mathbf{\Omega}_{\text{obs}})}{\lambda_{\min}(\mathbf{\Omega}_{\text{prior}}) + \beta(t)\lambda_{\min}(\mathbf{\Omega}_{\text{obs}})}.
\tag{30}
$$

Under Laplacian-regularized instantiations, $\lambda_{\min}(\mathbf{\Omega}_{\text{prior}}) \geq \varepsilon$ and $\lambda_{\min}(\mathbf{\Omega}_{\text{obs}}) \geq \varepsilon_{\text{obs}}$. When the coupling template keeps the maximum weighted degree of $\mathcal{H}$ bounded, $\lambda_{\max}(\mathcal{L})$ is controlled by that degree, which in turn prevents $\lambda_{\max}(\mathbf{\Omega}_{\text{prior}})$ and $\lambda_{\max}(\mathbf{\Omega}_{\text{obs}})$ from growing arbitrarily. In practice, we further apply a Jacobi preconditioner, diagonal scaling to reduce $\kappa(\boldsymbol{P}_t)$ and keep $K$ small (Benzi, 2002; Wathen, 2015). The wall-clock time and memory comparison is displayed in Appendix C.1.

**Proposition 4.5** (Positive definiteness of the update operator). *If $\mathbf{\Omega}_{\text{prior}} \succ \mathbf{0}$, $\mathbf{\Omega}_{\text{obs}} \succ \mathbf{0}$, and $\beta(t) \geq 0$, then $\boldsymbol{P}_t = \mathbf{\Omega}_{\text{prior}} + \beta(t)\mathbf{\Omega}_{\text{obs}} \succ \mathbf{0}$ for all $t \in [0, 1]$.*

*Table 1.* Performance comparison on Planar, Tree, and SBM datasets. The best results are highlighted in **bold** and the second best are underlined. Null values (-) in criteria indicate that statistics are unavailable from the original paper.

| Model | Class | Planar ($\|N\|=64$) | | Tree ($\|N\|=64$) | | SBM ($44\leq\|N\|\leq187$) | |
|---|---|---|---|---|---|---|---|
| | | V.U.N. ↑ | Ratio↓ | V.U.N. ↑ | Ratio↓ | V.U.N. ↑ | Ratio↓ |
| Training set | - | 100.0 | 1.0 | 100.0 | 1.0 | 100.0 | 1.0 |
| GraphRNN (You et al., 2018) | Autoregressive | 0.0 | 490.2 | 0.0 | 607.0 | 5.0 | 14.7 |
| GRAN (Liao et al., 2019) | Autoregressive | 0.0 | 2.0 | 0.0 | 607.0 | 25.0 | 9.7 |
| SPECTRE (Martinkus et al., 2022) | GAN | 25.0 | 3.0 | - | - | 52.5 | 2.2 |
| DiGress (Vignac et al., 2023) | Diffusion | 77.5 | 5.1 | 90.0 | 1.6 | 60.0 | 1.7 |
| EDGE (Chen et al., 2023) | Diffusion | 0.0 | 431.4 | 0.0 | 850.7 | 0.0 | 51.4 |
| BwR (Diamant et al., 2023) | Diffusion | 0.0 | 251.9 | 0.0 | 11.4 | 7.5 | 38.6 |
| BiGG (Dai et al., 2020) | Autoregressive | 5.0 | 16.0 | 75.0 | 5.2 | 10.0 | 11.9 |
| GraphGen (Goyal et al., 2020) | Autoregressive | 7.5 | 210.3 | 95.0 | 33.2 | 5.0 | 48.8 |
| HSpectre (Bergmeister et al., 2024) | Diffusion | 95.0 | 2.1 | **100.0** | 4.0 | 75.0 | 10.5 |
| GruM (Jo et al., 2024) | Diffusion | 90.0 | 1.8 | - | - | 85.0 | **1.1** |
| CatFlow (Eijkelboom et al., 2024) | Flow | 80.0 | - | - | - | 85.0 | - |
| DisCo (Xu et al., 2024) | Diffusion | 83.6 | - | - | - | 66.2 | - |
| Cometh (Siraudin et al., 2025) | Diffusion | **99.5** | - | - | - | 75.0 | - |
| SID (Boget, 2025) | Diffusion | 91.3 | - | - | - | 63.5 | - |
| DeFoG (Qin et al., 2025) | Flow | **99.5** | 1.6 | 96.5 | 1.6 | **90.0** | 4.9 |
| TopBF (Xiong et al., 2026) | Bayesian Flow | 98.3 | 1.8 | 95.2 | 1.7 | 89.2 | 3.3 |
| VBFN | Bayesian Flow | **99.5** | **1.5** | 97.3 | **1.4** | 89.7 | 1.5 |

**Gradients and memory.** The solve produces the belief state $\theta_t$ used as input to the predictor $\widehat{z}_\phi(\theta_t, t)$. In the objective Eq. (27), gradients are taken with respect to $\phi$ only. We treat $\theta_t$ as a sampled state generated by the fixed training mechanism and do not backpropagate through the solver, avoiding unrolling or implicit differentiation and keeping memory usage comparable to classical BFN training.

## 5. Experiments

### 5.1. Synthetic Graph Generation

**Datasets and metrics** We evaluate VBFN on three widely used synthetic graph generation datasets, including **Planar**, **Tree** (Bergmeister et al., 2024), and Stochastic Block Model (**SBM**) (Martinkus et al., 2022). We measure graph-level validity, uniqueness, and novelty percentage by **V.U.N.**, and the average ratio graph-theoretic distance of generated and test sets against that of train and test sets by **Ratio**, following (Qin et al., 2025).

**Results** Table 1 demonstrates that VBFN achieves a superior balance between graph validity and distributional fidelity, consistently ranking among the top two across all benchmarks. On Planar graphs, VBFN establishes the best Ratio and V.U.N., significantly outperforming other diffusion and flow-based baselines. On the Tree dataset, while HSpectre achieves perfect validity, VBFN attains a sub-

stantially better Ratio of 1.4 than 4.0, indicating that our coupled Bayesian update prevents the generation of valid but structurally trivial samples. This robustness extends to the complex SBM dataset, where VBFN secures the second-best results in both metrics, unlike previous methods that often suffer from a trade-off between V.U.N. and Ratio. VBFN consistently occupies the top tier, validating the efficacy of Bayesian flow in capturing high-order geometric dependencies.

### 5.2. Molecular Graph Generation

**Datasets and metrics** We conduct experiments on two standard benchmarks: QM9 (Ramakrishnan et al., 2014), consisting of small molecules with up to 9 heavy atoms labeled with quantum chemical properties, and ZINC250k (Irwin et al., 2012), a more challenging dataset containing 250k drug-like commercially available molecules with up to 38 atoms. Performance is assessed using four key metrics. **Valid** denotes the percentage of generated graphs that represent chemically valid molecules. **FCD** (Fréchet ChemNet Distance) (Preuer et al., 2018) measures the distance between generated and real distributions in the chemical feature space. Neighborhood Subgraph Pairwise Distance Kernel (**NSPDK**) maximum mean discrepancy (MMD) (Costa & De Grave, 2010) quantifies the structural discrepancy using the Neighborhood Subgraph Pairwise Distance Kernel. Scaffold similarity (**Scaf.**) (Bemis & Murcko, 1996) eval-

*Table 2.* Performance comparison on QM9 and ZINC250k datasets. The best results are highlighted in **bold** and the second best are underlined. Null values (-) in criteria indicate that statistics are unavailable from the original paper.

| Method | QM9 $(|N| \leq 9)$ | | | | ZINC250k $(|N| \leq 38)$ | | | |
|---|---|---|---|---|---|---|---|---|
| | Valid ↑ | FCD↓ | NSPDK↓ | Scaf.↑ | Valid ↑ | FCD↓ | NSPDK↓ | Scaf.↑ |
| Training set | 100.00 | 0.0398 | 0.0001 | 0.9717 | 100.00 | 0.0615 | 0.0001 | 0.8395 |
| GDSS (Jo et al., 2022) | 95.72 | 2.900 | 0.0033 | 0.6983 | 97.01 | 14.656 | 0.0195 | 0.0467 |
| DiGress (Vignac et al., 2023) | 98.19 | 0.095 | 0.0003 | 0.9353 | 94.99 | 3.482 | 0.0021 | 0.4163 |
| GSDM (Luo et al., 2023) | 99.90 | 2.614 | 0.0034 | - | 92.57 | 12.435 | 0.0168 | - |
| LGD-large (Zhou et al., 2024) | 99.13 | 0.104 | 0.0002 | - | - | - | - | - |
| GruM (Jo et al., 2024) | 99.69 | 0.108 | 0.0002 | 0.9449 | 98.65 | 2.257 | 0.0015 | 0.5299 |
| SID (Boget, 2025) | 99.67 | 0.504 | **0.0001** | - | 99.50 | 2.010 | 0.0021 | - |
| DeFoG (Qin et al., 2025) | 99.30 | 0.120 | - | - | 99.22 | 1.425 | 0.0008 | 0.5903 |
| TopBF (Xiong et al., 2026) | 99.74 | 0.093 | 0.0002 | 0.9421 | 99.37 | 1.392 | 0.0008 | 0.5372 |
| VBFN | **99.98** | **0.083** | 0.0002 | **0.9519** | **99.63** | **1.307** | **0.0007** | **0.6057** |

*Table 3.* QM9 ablations for VBFN. * denotes default setting.

| Setting | Valid ↑ | FCD ↓ | NSPDK ↓ | Scaf.↑ |
|---|---|---|---|---|
| $\lambda_X=0, \lambda_A=0$ | 99.51 | 25.926 | 1.1126 | 0.0 |
| $\lambda_X=1, \lambda_A=0$ | 0.01 | 16.951 | 0.2289 | 0.0 |
| $\lambda_X=0, \lambda_A=1$ | 51.34 | 17.171 | 0.0472 | 0.5884 |
| $\lambda_X=1, \lambda_A=1$* | 99.98 | 0.083 | 0.0002 | 0.9519 |
| $\mathbf{\Omega}_{obs}$: prior | 42.71 | 8.717 | 0.0402 | 0.1893 |
| $\mathbf{\Omega}_{obs}$: diag_prior* | 99.98 | 0.083 | 0.0002 | 0.9519 |
| Cholesky | 99.98 | 0.084 | 0.0002 | 0.9502 |
| CG* | 99.98 | 0.083 | 0.0002 | 0.9519 |

uates the diversity and realism of the generated molecular substructures compared to the training set.

**Results** As shown in Table 2, VBFN achieves state-of-the-art performance across both datasets. On QM9, VBFN reaches near-perfect validity by 99.98%, the lowest FCD by 0.083, and the best scaffold similarity by 0.9519, surpassing previous baselines. The advantages of VBFN are clearer on the larger and more complex ZINC250k dataset. VBFN not only achieves the highest validity by 99.63% but also reduces FCD to 1.307 and NSPDK to 0.0007. Crucially, VBFN attains the highest Scaffold similarity by 0.6057, indicating that our coupled Bayesian update mechanism effectively captures long-range dependencies and complex functional groups. This confirms VBFN can generate chemically plausible molecules with superior structural fidelity.

### 5.3. Ablation Studies

As shown in Table 3, we conduct three ablation studies on QM9 to assess the roles of precision-induced coupling, the observation-channel design, and the linear solver in VBFN. Detailed ablation studies are in Appendix C.2.

**Where coupling matters.** The full model ($\lambda_X = \lambda_A = 1$) achieves strong validity and better FCD/NSPDK with high scaffold similarity. In contrast, disabling coupling severely degrades global statistics in FCD/NSPDK and collapses scaffold similarity, despite high validity under $\lambda_X = \lambda_A = 0$. Single-sided coupling is not sufficient under the same configuration. Coupling only $A$ partially restores structural signals (notably Scaf.), whereas coupling only $X$ fails to produce valid molecules. Overall, these results support that chemically plausible graphs require joint propagation between node and edge degrees of freedom, which is realized in VBFN through the coupled Bayesian update.

**Prior-only versus structured channel.** Setting $\mathbf{\Omega}_{obs} = \mathbf{\Omega}_{prior}$ substantially hurts validity and alignment of using a diagonal observation precision. When $\mathbf{\Omega}_{obs} = \mathbf{\Omega}_{prior}$, we have $\mathbf{P}_t = (1 + \beta(t))\mathbf{\Omega}_{prior}$, so the structured precision cancels in the update and $\boldsymbol{\theta}_t$ reduces to a simple convex combination of $\boldsymbol{\theta}_0$ and $\boldsymbol{y}$, weakening prior-induced propagation. A diagonal channel avoids this collapse and injects evidence in a more stable, coordinate-local manner while preserving coupling through $\mathbf{\Omega}_{prior}$.

**Linear solver.** CG and Cholesky yield essentially identical generation metrics when solving the same SPD system to comparable accuracy, confirming that solver choice does not affect the learned generative behavior. We therefore use CG as the default implementation, while Cholesky remains a viable option for small graphs.

### 6. Conclusion

We introduced VBFN for graph generation by replacing the independent coding geometry of classical BFNs with a joint Gaussian variational belief governed by structured precisions. By building a representation-induced dependency

graph without querying authentic adjacency values, VBFN instantiates Laplacian-based precisions coupling node and edge variables while avoiding label leakage. Under tied message covariance, we retain the classical message–matching objective, but the belief evolution is induced by a joint Bayesian update that reduces to solving an SPD linear system at each time step. Empirically, VBFN improves fidelity and structural coherence across synthetic and molecular graphs, supporting the role of precision-induced coupling as an intrinsic mechanism for geometric belief propagation.

## Acknowledgement

This work was supported in part by the National Key Research and Development Program of China (2023YFC2705705), the Natural Science Foundation of China (No.62476203), Key Project of Traditional Chinese Medicine Joint Fund of Hubei Provincial Natural Science Foundation (No.2025AFD47), Hubei Province Science and Technology Innovation Plan Project (No.2025BCB035), the Guangdong Provincial Natural Science Foundation General Project (No.2025A1515012155), the Shenzhen Natural Science Foundation Project (No. JCYJ20250604122534006, JCYJ20230807090211021), the Hubei Jiangxia laboratory development project (No.2025JXKF034), the New Cornerstone Science Foundation through the XPLORER PRIZE, and the Innovative Research Group Project of Hubei Province under Grants 2024AFA017.

## Impact Statement

This paper develops a variational Bayesian flow framework for generating graph-structured data with coupled node–edge updates. The main intended impact is methodological, and we expect it to support downstream scientific applications that rely on discrete graph generation, including molecular design in drug discovery. Better generative models can help narrow the search space of candidate molecules and prioritize compounds for subsequent computational screening, which may reduce experimental cost and shorten iteration cycles. At the same time, any improvement in molecular generation can be dual-use. Inappropriate use could include generating candidates that are harmful to humans or the environment if combined with additional domain knowledge and synthesis-oriented tooling. Our work does not provide synthesis instructions or target–specific hazardous objectives, and we encourage responsible use with standard screening and access controls in downstream pipelines.

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

# A. Proofs and Derivations

## A.1. Proof of Theorem 4.1

*Proof.* By Bayesian update,

$$p(\boldsymbol{z} \mid \boldsymbol{y}; t) \propto p_{\text{prior}}(\boldsymbol{z}) \, p_{\text{S}}(\boldsymbol{y} \mid \boldsymbol{z}; t). \tag{31}$$

Under the theorem assumptions, $p_{\text{prior}}(\boldsymbol{z}) = \mathcal{N}(\boldsymbol{z}; \boldsymbol{\theta}_0, \boldsymbol{\Omega}_{\text{prior}}^{-1})$ and $p_{\text{S}}(\boldsymbol{y} \mid \boldsymbol{z}; t) = \mathcal{N}(\boldsymbol{y}; \boldsymbol{z}, (\beta(t)\boldsymbol{\Omega}_{\text{obs}})^{-1})$. Ignoring additive constants independent of $\boldsymbol{z}$, their log-densities as functions of $\boldsymbol{z}$ are

$$\log p_{\text{prior}}(\boldsymbol{z}) = -\tfrac{1}{2}(\boldsymbol{z} - \boldsymbol{\theta}_0)^{\top}\boldsymbol{\Omega}_{\text{prior}}(\boldsymbol{z} - \boldsymbol{\theta}_0) + \text{const}, \tag{32}$$

$$\log p_{\text{S}}(\boldsymbol{y} \mid \boldsymbol{z}; t) = -\tfrac{1}{2}(\boldsymbol{y} - \boldsymbol{z})^{\top}\big(\beta(t)\boldsymbol{\Omega}_{\text{obs}}\big)(\boldsymbol{y} - \boldsymbol{z}) + \text{const}. \tag{33}$$

Summing Eq. (32) and Eq. (33) and expanding all quadratic terms in $\boldsymbol{z}$ yields

$$\log p(\boldsymbol{z} \mid \boldsymbol{y}; t) = -\tfrac{1}{2}\boldsymbol{z}^{\top}\boldsymbol{P}_t\boldsymbol{z} + \boldsymbol{z}^{\top}\boldsymbol{h}_t + \text{const}, \tag{34}$$

where

$$\boldsymbol{P}_t \triangleq \boldsymbol{\Omega}_{\text{prior}} + \beta(t)\boldsymbol{\Omega}_{\text{obs}}, \qquad \boldsymbol{h}_t \triangleq \boldsymbol{\Omega}_{\text{prior}}\boldsymbol{\theta}_0 + \beta(t)\boldsymbol{\Omega}_{\text{obs}}\boldsymbol{y}. \tag{35}$$

Since $\boldsymbol{P}_t \succ \boldsymbol{0}$ (Proposition 4.5), define $\boldsymbol{\theta}_t \triangleq \boldsymbol{P}_t^{-1}\boldsymbol{h}_t$.

Now rewrite the quadratic form by shifting $\boldsymbol{z}$ around $\boldsymbol{\theta}_t$. Let $\boldsymbol{r} \triangleq \boldsymbol{z} - \boldsymbol{\theta}_t$ (equivalently $\boldsymbol{z} = \boldsymbol{r} + \boldsymbol{\theta}_t$). Then

$$-\tfrac{1}{2}\boldsymbol{z}^{\top}\boldsymbol{P}_t\boldsymbol{z} + \boldsymbol{z}^{\top}\boldsymbol{h}_t = -\tfrac{1}{2}(\boldsymbol{r} + \boldsymbol{\theta}_t)^{\top}\boldsymbol{P}_t(\boldsymbol{r} + \boldsymbol{\theta}_t) + (\boldsymbol{r} + \boldsymbol{\theta}_t)^{\top}\boldsymbol{h}_t \tag{36}$$

$$= -\tfrac{1}{2}\boldsymbol{r}^{\top}\boldsymbol{P}_t\boldsymbol{r} - \boldsymbol{r}^{\top}\boldsymbol{P}_t\boldsymbol{\theta}_t - \tfrac{1}{2}\boldsymbol{\theta}_t^{\top}\boldsymbol{P}_t\boldsymbol{\theta}_t + \boldsymbol{r}^{\top}\boldsymbol{h}_t + \boldsymbol{\theta}_t^{\top}\boldsymbol{h}_t. \tag{37}$$

Because $\boldsymbol{P}_t\boldsymbol{\theta}_t = \boldsymbol{h}_t$, the two terms linear in $\boldsymbol{r}$ cancel:

$$-\boldsymbol{r}^{\top}\boldsymbol{P}_t\boldsymbol{\theta}_t + \boldsymbol{r}^{\top}\boldsymbol{h}_t = -\boldsymbol{r}^{\top}\boldsymbol{h}_t + \boldsymbol{r}^{\top}\boldsymbol{h}_t = 0, \tag{38}$$

and also $\boldsymbol{\theta}_t^{\top}\boldsymbol{h}_t = \boldsymbol{\theta}_t^{\top}\boldsymbol{P}_t\boldsymbol{\theta}_t$. Therefore Eq. (37) becomes

$$-\tfrac{1}{2}\boldsymbol{z}^{\top}\boldsymbol{P}_t\boldsymbol{z} + \boldsymbol{z}^{\top}\boldsymbol{h}_t = -\tfrac{1}{2}(\boldsymbol{z} - \boldsymbol{\theta}_t)^{\top}\boldsymbol{P}_t(\boldsymbol{z} - \boldsymbol{\theta}_t) + \tfrac{1}{2}\boldsymbol{\theta}_t^{\top}\boldsymbol{P}_t\boldsymbol{\theta}_t. \tag{39}$$

The last term is constant with respect to $\boldsymbol{z}$, hence

$$p(\boldsymbol{z} \mid \boldsymbol{y}; t) \propto \exp\Big(-\tfrac{1}{2}(\boldsymbol{z} - \boldsymbol{\theta}_t)^{\top}\boldsymbol{P}_t(\boldsymbol{z} - \boldsymbol{\theta}_t)\Big). \tag{40}$$

This is exactly the Gaussian kernel with mean $\boldsymbol{\theta}_t$ and covariance $\boldsymbol{P}_t^{-1}$, so

$$p(\boldsymbol{z} \mid \boldsymbol{y}; t) = \mathcal{N}(\boldsymbol{z}; \boldsymbol{\theta}_t, \boldsymbol{P}_t^{-1}). \tag{41}$$

Finally, multiplying $\boldsymbol{\theta}_t = \boldsymbol{P}_t^{-1}\boldsymbol{h}_t$ by $\boldsymbol{P}_t$ gives the linear system form in Theorem 4.1, completing the proof. $\square$

## A.2. Proof of Proposition 4.2

*Proof.* Assume $\boldsymbol{\Omega}_{\text{prior}} = \text{diag}(\boldsymbol{\omega}_{\text{prior}})$ and $\boldsymbol{\Omega}_{\text{obs}} = \text{diag}(\boldsymbol{\omega}_{\text{obs}})$. Then

$$\boldsymbol{P}_t = \boldsymbol{\Omega}_{\text{prior}} + \beta(t)\boldsymbol{\Omega}_{\text{obs}} = \text{diag}\big(\omega_{\text{prior},1} + \beta(t)\omega_{\text{obs},1}, \ldots, \omega_{\text{prior},D} + \beta(t)\omega_{\text{obs},D}\big) \tag{42}$$

is diagonal, hence invertible element-wise. Theorem 4.1 gives

$$\boldsymbol{P}_t\boldsymbol{\theta}_t = \boldsymbol{\Omega}_{\text{prior}}\boldsymbol{\theta}_0 + \beta(t)\boldsymbol{\Omega}_{\text{obs}}\boldsymbol{y}. \tag{43}$$

Taking the $i$-th entry of Eq. (43) yields the scalar update rule

$$\big(\omega_{\text{prior},i} + \beta(t)\omega_{\text{obs},i}\big)\theta_{t,i} = \omega_{\text{prior},i}\theta_{0,i} + \beta(t)\omega_{\text{obs},i}y_i, \tag{44}$$

so

$$\theta_{t,i} = \frac{\omega_{\text{prior},i}\theta_{0,i} + \beta(t)\omega_{\text{obs},i}y_i}{\omega_{\text{prior},i} + \beta(t)\omega_{\text{obs},i}}. \tag{45}$$

This is exactly index-wise Gaussian conjugate fusion: each dimension is updated independently using a weighted average whose weights are precisions. When $\boldsymbol{\omega}_{\text{obs}} = \boldsymbol{1}$ and $\boldsymbol{\omega}_{\text{prior}}$ follows the classical BFN schedule, Eq. (45) recovers the classical factorized-Gaussian BFN update. $\square$

### A.3. Proof of Proposition 4.3

*Proof.* Define the objective

$$J(\boldsymbol{u}) = \frac{1}{2}(\boldsymbol{u} - \boldsymbol{\theta}_0)^\top \boldsymbol{\Omega}_{\text{prior}}(\boldsymbol{u} - \boldsymbol{\theta}_0) + \frac{\beta(t)}{2}(\boldsymbol{y} - \boldsymbol{u})^\top \boldsymbol{\Omega}_{\text{obs}}(\boldsymbol{y} - \boldsymbol{u}). \tag{46}$$

Expanding the two quadratic forms gives

$$J(\boldsymbol{u}) = \frac{1}{2}\boldsymbol{u}^\top \boldsymbol{\Omega}_{\text{prior}}\boldsymbol{u} - \boldsymbol{u}^\top \boldsymbol{\Omega}_{\text{prior}}\boldsymbol{\theta}_0 + \frac{1}{2}\boldsymbol{\theta}_0^\top \boldsymbol{\Omega}_{\text{prior}}\boldsymbol{\theta}_0 + \frac{\beta(t)}{2}\boldsymbol{u}^\top \boldsymbol{\Omega}_{\text{obs}}\boldsymbol{u} - \beta(t)\boldsymbol{u}^\top \boldsymbol{\Omega}_{\text{obs}}\boldsymbol{y} + \frac{\beta(t)}{2}\boldsymbol{y}^\top \boldsymbol{\Omega}_{\text{obs}}\boldsymbol{y}. \tag{47}$$

Differentiating with respect to $\boldsymbol{u}$ and using symmetry of $\boldsymbol{\Omega}_{\text{prior}}, \boldsymbol{\Omega}_{\text{obs}}$ yields

$$\nabla_{\boldsymbol{u}} J(\boldsymbol{u}) = \boldsymbol{\Omega}_{\text{prior}}\boldsymbol{u} - \boldsymbol{\Omega}_{\text{prior}}\boldsymbol{\theta}_0 + \beta(t)\boldsymbol{\Omega}_{\text{obs}}\boldsymbol{u} - \beta(t)\boldsymbol{\Omega}_{\text{obs}}\boldsymbol{y}. \tag{48}$$

Setting $\nabla_{\boldsymbol{u}} J(\boldsymbol{u}) = \boldsymbol{0}$ gives

$$\big(\boldsymbol{\Omega}_{\text{prior}} + \beta(t)\boldsymbol{\Omega}_{\text{obs}}\big)\boldsymbol{u} = \boldsymbol{\Omega}_{\text{prior}}\boldsymbol{\theta}_0 + \beta(t)\boldsymbol{\Omega}_{\text{obs}}\boldsymbol{y}, \tag{49}$$

which is exactly the linear system in Theorem 4.1. Since $\boldsymbol{\Omega}_{\text{prior}} + \beta(t)\boldsymbol{\Omega}_{\text{obs}} \succ \boldsymbol{0}$ (Proposition 4.5), $J$ is strictly convex and the minimizer is unique; it equals the posterior belief parameter $\boldsymbol{\theta}_t$. $\qquad\square$

### A.4. Proof of Proposition 4.4

*Proof.* Let

$$p_{\text{S}}(\boldsymbol{y}) = \mathcal{N}(\boldsymbol{y}; \boldsymbol{m}_1, \boldsymbol{\Sigma}), \qquad p_{\text{R}}(\boldsymbol{y}) = \mathcal{N}(\boldsymbol{y}; \boldsymbol{m}_2, \boldsymbol{\Sigma}),$$

with a common covariance $\boldsymbol{\Sigma} \succ \boldsymbol{0}$. The KL divergence between two Gaussians is

$$D_{\text{KL}}\big(\mathcal{N}(\boldsymbol{m}_1, \boldsymbol{\Sigma}) \,\big\|\, \mathcal{N}(\boldsymbol{m}_2, \boldsymbol{\Sigma})\big) = \frac{1}{2}\Big[\text{tr}(\boldsymbol{\Sigma}^{-1}\boldsymbol{\Sigma}) + (\boldsymbol{m}_2 - \boldsymbol{m}_1)^\top \boldsymbol{\Sigma}^{-1}(\boldsymbol{m}_2 - \boldsymbol{m}_1) - D + \log \frac{\det \boldsymbol{\Sigma}}{\det \boldsymbol{\Sigma}}\Big]. \tag{50}$$

Now, $\text{tr}(\boldsymbol{\Sigma}^{-1}\boldsymbol{\Sigma}) = \text{tr}(\boldsymbol{I}) = D$, and $\log(\det \boldsymbol{\Sigma} / \det \boldsymbol{\Sigma}) = 0$. Hence

$$D_{\text{KL}}\big(\mathcal{N}(\boldsymbol{m}_1, \boldsymbol{\Sigma}) \,\big\|\, \mathcal{N}(\boldsymbol{m}_2, \boldsymbol{\Sigma})\big) = \frac{1}{2}(\boldsymbol{m}_2 - \boldsymbol{m}_1)^\top \boldsymbol{\Sigma}^{-1}(\boldsymbol{m}_2 - \boldsymbol{m}_1). \tag{51}$$

In VBFN, Proposition 4.4 assumes tied channel covariance $\boldsymbol{\Sigma} = (\beta(t)\boldsymbol{\Omega}_{\text{obs}})^{-1}$, and the message means $\boldsymbol{m}_1 = \boldsymbol{z}$, $\boldsymbol{m}_2 = \widehat{\boldsymbol{z}}_\phi(\boldsymbol{\theta}_t, t)$. Substituting into Eq. (51) gives

$$D_{\text{KL}}\big(p_{\text{S}}(\cdot \mid \boldsymbol{z}; t) \,\big\|\, p_{\text{R}}(\cdot \mid \boldsymbol{\theta}_t; t)\big) = \frac{1}{2}\big(\widehat{\boldsymbol{z}}_\phi(\boldsymbol{\theta}_t, t) - \boldsymbol{z}\big)^\top \big(\beta(t)\boldsymbol{\Omega}_{\text{obs}}\big)\big(\widehat{\boldsymbol{z}}_\phi(\boldsymbol{\theta}_t, t) - \boldsymbol{z}\big) \tag{52}$$

$$= \frac{\beta(t)}{2}\big\|\boldsymbol{z} - \widehat{\boldsymbol{z}}_\phi(\boldsymbol{\theta}_t, t)\big\|_{\boldsymbol{\Omega}_{\text{obs}}}^2. \tag{53}$$

which is exactly the weighted quadratic form stated in the proposition. $\qquad\square$

### A.5. Proof of Proposition 4.5

*Proof.* For any nonzero $\boldsymbol{v} \in \mathbb{R}^D$,

$$\boldsymbol{v}^\top \big(\boldsymbol{\Omega}_{\text{prior}} + \beta(t)\boldsymbol{\Omega}_{\text{obs}}\big)\boldsymbol{v} = \boldsymbol{v}^\top \boldsymbol{\Omega}_{\text{prior}}\boldsymbol{v} + \beta(t)\,\boldsymbol{v}^\top \boldsymbol{\Omega}_{\text{obs}}\boldsymbol{v}. \tag{54}$$

Since $\boldsymbol{\Omega}_{\text{prior}} \succ \boldsymbol{0}$, we have $\boldsymbol{v}^\top \boldsymbol{\Omega}_{\text{prior}}\boldsymbol{v} > 0$. Since $\boldsymbol{\Omega}_{\text{obs}} \succ \boldsymbol{0}$ and $\beta(t) \geq 0$, the second term is nonnegative. Therefore, the sum is strictly positive for all $\boldsymbol{v} \neq \boldsymbol{0}$, proving the matrix is positive definite. $\qquad\square$

### A.6. Discrete-to-continuous Derivation of $\mathcal{L}_\infty$

We derive the continuous-time objective as the limit of the discrete-time message–matching ELBO.

**Discrete partition.** Let $0 = t_0 < t_1 < \cdots < t_T = 1$ be a partition with $\Delta t_k = t_k - t_{k-1}$. Define $\Delta \beta_k = \beta(t_k) - \beta(t_{k-1})$. For sufficiently fine partitions, $\Delta \beta_k = \alpha(\xi_k)\Delta t_k$ for some $\xi_k \in (t_{k-1}, t_k)$ by the mean-value theorem.

**Stepwise KL.** At step $k$, the flow objective weights the KL at time $t_{k-1}$ by $\Delta\beta_k$. Under tied covariance $(\beta(t_{k-1})\mathbf{\Omega}_{\mathrm{obs}})^{-1}$, Proposition 4.4 gives

$$D_{\mathrm{KL}}^{(k)} = \frac{\beta(t_{k-1})}{2} \left\| \boldsymbol{z} - \widehat{\boldsymbol{z}}_\phi(\boldsymbol{\theta}_{t_{k-1}}, t_{k-1}) \right\|_{\mathbf{\Omega}_{\mathrm{obs}}}^2 . \tag{55}$$

**Discrete objective.** Summing along the flow yields the discrete-time objective

$$\mathcal{L}_T(\phi) = \mathbb{E}_{\boldsymbol{z} \sim p_{\mathrm{data}}} \, \mathbb{E}_{\boldsymbol{\theta}_{t_{0:T}} \sim p_F(\cdot|\boldsymbol{z})} \left[ \sum_{k=1}^T \frac{\beta(t_{k-1})\,\Delta\beta_k}{2} \left\| \boldsymbol{z} - \widehat{\boldsymbol{z}}_\phi(\boldsymbol{\theta}_{t_{k-1}}, t_{k-1}) \right\|_{\mathbf{\Omega}_{\mathrm{obs}}}^2 \right]. \tag{56}$$

**Riemann limit.** As $\max_k \Delta t_k \to 0$, we have $\Delta\beta_k = \alpha(\xi_k)\Delta t_k$, and the sum in Eq. (56) converges (in the Riemann–Stieltjes sense) to

$$\mathcal{L}_\infty(\phi) = \mathbb{E}_{\boldsymbol{z} \sim p_{\mathrm{data}}} \int_0^1 \mathbb{E}_{\boldsymbol{\theta}_t \sim p_F(\cdot|\boldsymbol{z};t)} \left[ \frac{\alpha(t)\beta(t)}{2} \left\| \boldsymbol{z} - \widehat{\boldsymbol{z}}_\phi(\boldsymbol{\theta}_t, t) \right\|_{\mathbf{\Omega}_{\mathrm{obs}}}^2 \right] dt, \tag{57}$$

which matches Eq. (27) in the main text.

## B. Implementation Details

### B.1. Continuous Parameterization and Constraints

To train VBFN on graphs with *discrete* node/edge attributes, we first apply a continuous parameterization that preserves differentiability while keeping decoding simple. Specifically, each graph $G = (X, A)$ is mapped to a continuous relaxation $g(G)$, and we vectorize it as

$$\boldsymbol{z} \triangleq \mathrm{vec}\big(g(G)\big) \in \mathbb{R}^D. \tag{58}$$

For variable-size graphs, we use a binary mask $\boldsymbol{m} \in \{0,1\}^D$ and the diagonal masking operator $\boldsymbol{M} \triangleq \mathrm{diag}(\boldsymbol{m})$ to zero invalid indices and prevent spurious couplings.

**Class-to-continuous mapping.** For any discrete attribute with $K$ classes, we associate each class $k \in \{1, \ldots, K\}$ with a center value $k_c \in [-1, 1]$ and its left/right boundaries $(k_l, k_r)$:

$$\begin{aligned}
k_c &= \frac{2k - 1}{K} - 1, \\
k_l &= k_c - \frac{1}{K}, \\
k_r &= k_c + \frac{1}{K}.
\end{aligned} \tag{59}$$

We encode the $k$-th class by its continuous center $x_k := k_c$, yielding a continuous target representation $\mathbf{G} = (\mathbf{X}, \mathbf{A}) = g(G)$. During training and sampling, the model predicts Gaussian parameters $(\boldsymbol{\mu}, \boldsymbol{\sigma})$ over these continuous targets; the probability of each discrete class is obtained by integrating the Gaussian mass over the corresponding class interval $[k_l, k_r]$ via a truncated Gaussian cumulative distribution function (CDF), and a continuous center estimate can be formed as the expectation over class centers.

**Decoding and constraints.** The inverse mapping $g^{-1}$ converts continuous predictions back to discrete attributes and enforces structural constraints through simple projection operators. Examples include rounding to the nearest class center for discrete attributes, enforcing symmetry for undirected adjacency, and applying padding conventions under $\boldsymbol{m}$. These operations affect only the parameterization layer and are orthogonal to the core contribution of VBFN, which lies in the joint precision geometry and the induced Bayesian update.

### B.2. Truncated Gaussian CDF and per-class Probabilities.

VBFN represents discrete attributes by continuous class centers in $[-1, 1]$. At time $t$, the backbone $\widehat{\boldsymbol{z}}_\phi$ predicts for each discrete scalar attribute a Gaussian parameter pair $(\mu^{(d)}, \sigma^{(d)})$ with $\sigma^{(d)} > 0$, which we convert into a categorical distribution

over $K$ classes by integrating the Gaussian mass over class intervals. This CDF-based decoding is used for the edge channel on all datasets, and for the node channel whenever node attributes are discrete. For continuous node attributes, we directly regress $X$ and skip this decoding.

For a scalar attribute dimension $d$, define the Gaussian CDF

$$F(x \mid \mu^{(d)}, \sigma^{(d)}) = \frac{1}{2}\left[1 + \text{erf}\left(\frac{x - \mu^{(d)}}{\sqrt{2}\,\sigma^{(d)}}\right)\right], \tag{60}$$

where $\text{erf}(u) = \frac{2}{\sqrt{\pi}}\int_0^u e^{-t^2}\,dt$. Since our class centers and boundaries lie in $[-1, 1]$, we use the truncated CDF

$$\mathcal{F}(x \mid \mu^{(d)}, \sigma^{(d)}) = \begin{cases} 0, & x \leq -1, \\ 1, & x \geq 1, \\ F(x \mid \mu^{(d)}, \sigma^{(d)}), & \text{otherwise.} \end{cases} \tag{61}$$

Given class boundaries $\{(k_l, k_r)\}_{k=1}^{K}$ from Eq. (59), the probability mass assigned to class $k$ is

$$p^{(d)}(k) = \mathcal{F}(k_r \mid \mu^{(d)}, \sigma^{(d)}) - \mathcal{F}(k_l \mid \mu^{(d)}, \sigma^{(d)}), \qquad k = 1, \ldots, K. \tag{62}$$

We apply validity masks and renormalize over classes so that $\sum_k p^{(d)}(k) = 1$ on valid entries.

During training, we use the expected class-center representation:

$$\widehat{x}_c^{(d)} = \sum_{k=1}^{K} p^{(d)}(k)\, k_c, \tag{63}$$

and compute masked MSE between $\widehat{x}_c^{(d)}$ and the ground-truth class center $x_c^{(d)}$. During sampling, we decode discretely by either $\arg\max_k p^{(d)}(k)$ or by sampling from the categorical distribution $p^{(d)}(\cdot)$, followed by dataset-specific projection steps.

### B.3. Construction of the Observation Precision $\mathbf{\Omega}_{\text{obs}}$

This appendix specifies how we instantiate the observation precision $\mathbf{\Omega}_{\text{obs}}$ used in the structured channel (Eqs (17) and (18)). Recall that we require $\mathbf{\Omega}_{\text{obs}} \succ \mathbf{0}$ so that $\|\cdot\|_{\mathbf{\Omega}_{\text{obs}}}$ is a true norm and the quadratic in Proposition 4.4 is non-degenerate.

**Laplacian-based construction and SPD regularization.** We reuse the same fixed dependency graph $\mathcal{H} = (\mathcal{V}_{\mathcal{H}}, \mathcal{E}_{\mathcal{H}})$ introduced for the prior in Sec. 4.2. Let $\mathcal{L}$ denote its weighted combinatorial Laplacian as defined in Eq. (14). Since $\mathcal{L}$ is positive semidefinite, it may have a non-trivial null space. To avoid collapse along this null space, we add a diagonal regularization term, mirroring the prior construction in Eq. (15).

Concretely, given a diagonal mask operator $M$ for padding/validity constraints, we define the Laplacian-based observation precision as

$$\mathbf{\Omega}_{\text{obs}} = M\mathcal{L}M + \varepsilon_{\text{obs}}I, \qquad \varepsilon_{\text{obs}} > 0. \tag{64}$$

The coupling strengths $(\lambda_X, \lambda_A)$ are encoded in the edge weights $\{w_{uv}\}$ and therefore in $\mathcal{L}$, so we do not introduce an additional scalar multiplier for $\mathbf{\Omega}_{\text{obs}}$. Placing $\varepsilon_{\text{obs}}I$ outside the masking guarantees strict positive definiteness even when $M$ zeroes invalid entries. Indeed, for any nonzero $v \in \mathbb{R}^D$,

$$v^\top \mathbf{\Omega}_{\text{obs}} v = (Mv)^\top \mathcal{L}(Mv) + \varepsilon_{\text{obs}}\|v\|_2^2 \geq \varepsilon_{\text{obs}}\|v\|_2^2 > 0, \tag{65}$$

where we used $\mathcal{L} \succeq \mathbf{0}$. Therefore $\mathbf{\Omega}_{\text{obs}} \succ \mathbf{0}$, and the induced quadratic $\|\cdot\|_{\mathbf{\Omega}_{\text{obs}}}^2$ is a valid norm.

**Diagonalized and independent variants.** In practice, we also consider simplified observation precisions derived from Eq. (64). A diagonalized variant keeps heterogeneous per-dimension accuracies while removing off-diagonal coupling

$$\mathbf{\Omega}_{\text{obs}} = \text{diag}(\mathbf{\Omega}_{\text{obs}}^{\text{lap}}), \qquad \mathbf{\Omega}_{\text{obs}}^{\text{lap}} \text{ given by Eq. (64)}, \tag{66}$$

and an independent variant sets $\mathbf{\Omega}_{\text{obs}} = I$. All choices satisfy $\mathbf{\Omega}_{\text{obs}} \succ \mathbf{0}$ and are compatible with the tied-covariance KL simplification in Proposition 4.4. The main text remains agnostic to which option is used, as the theoretical derivations only require strict positive definiteness and a fixed (sample-agnostic) operator.

*Table 4.* **Wall-clock time and memory on QM9 and ZINC250k.** We report average time per training epoch and sampling step, as well as peak GPU memory for training and sampling separately.

| Method | QM9 | | | | ZINC250k | | | |
| | Training | | Sampling | | Training | | Sampling | |
| | Time (s) | Mem (GB) | Time (s) | Mem (GB) | Time (s) | Mem (GB) | Time (s) | Mem (GB) |
|---|---|---|---|---|---|---|---|---|
| GruM | 57.91 | 8.55 | 0.82 | 6.80 | 793.05 | 26.28 | 3.02 | 24.80 |
| TopBF | 50.29 | 9.50 | 0.81 | 6.64 | 2107.63 | 22.73 | 2.97 | 24.66 |
| VBFN (Cholesky) | 36.52 | 8.60 | 0.81 | 6.95 | 845.07 | 27.37 | 3.25 | 45.38 |
| VBFN (CG) | 41.77 | 8.57 | 0.83 | 6.92 | 798.19 | 27.37 | 3.04 | 45.35 |

*Table 5.* Stage-wise allocated GPU memory in a profiled ZINC250k CG sampling run.

| Stage / tensor | Allocated Memory (GB) |
|---|---|
| After node precision construction | 0.07 |
| After edge precision construction | 9.28 |
| $\Omega^A_{\text{prior}}$ | 4.60 |
| $\Omega^A_{\text{obs}}$ | 4.60 |
| After posterior-cache init | 13.91 |
| $P_A$ | 4.60 |
| CG solve peak | 14.03 |
| Bayesian fusion peak | 23.18 |
| Predictor / decoder peak | 34.27 |

# C. Supplementary Experiments

While VBFN is applicable to both molecular and synthetic graph benchmarks, the commonly used synthetic datasets, Planar, SBM, and Tree, are small-scale with 200 graphs and thus less informative for evaluating efficiency and scalability. We therefore conduct most supplementary experiments on the molecular benchmarks QM9 and ZINC250k with more than 100,000 graphs, which better stress both the coupled Bayesian update and the linear solver used in VBFN.

## C.1. Wall-clock Time and Memory: Training and Sampling

We benchmark wall-clock time and peak memory against GruM and TopBF as representative and competitive Diffusion and BFN baselines with the same Graph Transformer backbone. For training, we report average time per epoch; for sampling, we report average time per sampling step. We also report peak GPU memory for training and sampling separately. All results are averaged over 5 runs with identical batch sizes (Training: 1,024 for QM9, 256 for ZINC250k; Sampling: 10,000 for QM9, 2,500 for ZINC250k) and sampling steps.

Table 4 summarizes wall-clock time and peak GPU memory on QM9 and ZINC250k. On QM9, VBFN is consistently faster than GruM in training, while memory remains essentially unchanged. Sampling on QM9 is also comparable across methods: time per step differs marginally, and peak memory stays within a narrow range. This indicates that for small graphs, the coupled precision update does not introduce a noticeable system overhead, and end-to-end cost is dominated by the shared neural backbone.

On ZINC250k, the efficiency picture changes due to larger coupled systems. Cholesky becomes less favorable, since it is slower in training and substantially increases sampling memory (45.38GB vs. 24.80GB for GruM), reflecting the higher memory footprint of factorization-related buffers at scale. CG avoids explicit factorization and yields a more stable profile: its training time and sampling time are both close to GruM, although sampling memory remains high, suggesting that the dominant bottleneck is not only factorization but also how coupled edge variables and precision-related caches are materialized during sampling. Overall, these results motivate CG as the default solver for large graphs and point to memory-efficient sampling as an orthogonal optimization direction.

To clarify what dominates the sampling memory cost in practice, we added a stage-wise memory profile for ZINC250k

*Table 6.* ZINC250k ablations for VBFN. * denotes default setting.

| Setting | Valid ↑ | FCD ↓ | NSPDK ↓ | Scaf.↑ |
|---|---|---|---|---|
| $\lambda_X=0, \lambda_A=0$ | 99.99 | 42.607 | 0.9502 | 0.0 |
| $\lambda_X=0.2, \lambda_A=0$ | 0.06 | 43.440 | 0.5279 | 0.0 |
| $\lambda_X=0, \lambda_A=0.2$ | 50.89 | 24.042 | 0.0435 | 0.0031 |
| $\lambda_X=0.2, \lambda_A=0.2$* | 99.63 | 1.307 | 0.0007 | 0.6057 |
| $\boldsymbol{\Omega}_{\text{obs}}$: prior | 51.22 | 21.350 | 0.0442 | 0.0026 |
| $\boldsymbol{\Omega}_{\text{obs}}$: diag_prior* | 99.63 | 1.307 | 0.0007 | 0.6057 |
| Cholesky | 99.50 | 1.211 | 0.0007 | 0.6049 |
| CG* | 99.63 | 1.307 | 0.0007 | 0.6057 |

*Table 7.* Random class-order permutation tests on QM9 and ZINC250k.

| | Ordering | Valid ↑ | FCD↓ | NSPDK↓ | Scaf.↑ |
|---|---|---|---|---|---|
| | C N O F (Default) | **99.98** | 0.083 | **0.0002** | **0.9519** |
| QM9 | N O F C | 99.96 | 0.087 | 0.0003 | 0.9481 |
| | F C O N | **99.98** | **0.081** | **0.0002** | 0.9500 |
| | C N O F P S Cl Br I (Default) | 99.63 | 1.307 | **0.0007** | **0.6057** |
| ZINC250k | S Br N C F I P Cl O | 99.61 | **1.303** | **0.0007** | 0.5991 |
| | P I C Cl F N S Br O | **99.67** | 1.389 | 0.0008 | 0.5716 |

sampling with the CG solver. As shown in Table 5, node-space costs are negligible, whereas constructing the dense edge-space precisions $\Omega^A_{\text{prior}}$ and $\Omega^A_{\text{obs}}$ already raises allocated memory from 0.07 GB to 9.28 GB, and initializing the edge posterior cache $P_A$ further raises it to 13.91 GB. By contrast, the CG solve itself peaks at only 14.03 GB. Thus, the main persistent bottleneck comes from the dense edge-space coupled geometry and cached posterior states, while the largest transient peaks arise in prediction/decoding and Bayes fusion.

### C.2. Ablation Studies on ZINC250k

Table 6 studies how coupling strengths and observation precision choices affect ZINC250k generation. The no-coupling setting $\lambda_X=0, \lambda_A=0$ attains 99.99% validity, but in practice collapses to generating repetitive, trivial molecules. The severe degradation in FCD (42.607), NSPDK (0.9502), and scaffold score (0.0) confirms this failure mode. Partial coupling is also inadequate. Node-only coupling ($\lambda_X=0.2, \lambda_A=0$) catastrophically breaks validity (0.06%), while edge-only coupling ($\lambda_X=0, \lambda_A=0.2$) improves distributional metrics but still yields low validity (50.89%) and near-zero scaffold diversity.

Full coupling ($\lambda_X=0.2, \lambda_A=0.2$) is required to simultaneously achieve high validity and distributional matching, supporting the hypothesis that joint node–edge precision geometry is central to VBFN. The observation precision ablation further shows that tying $\boldsymbol{\Omega}_{\text{obs}}$ to the prior is harmful on ZINC250k (Valid: 51.22, FCD: 21.350), whereas the default diagonalized observation precision is crucial for stable, high-quality generation. Finally, Cholesky and CG produce very similar generation metrics under the same geometry, indicating that solver choice primarily affects efficiency (Table 4) rather than sample quality when CG is run to sufficient accuracy.

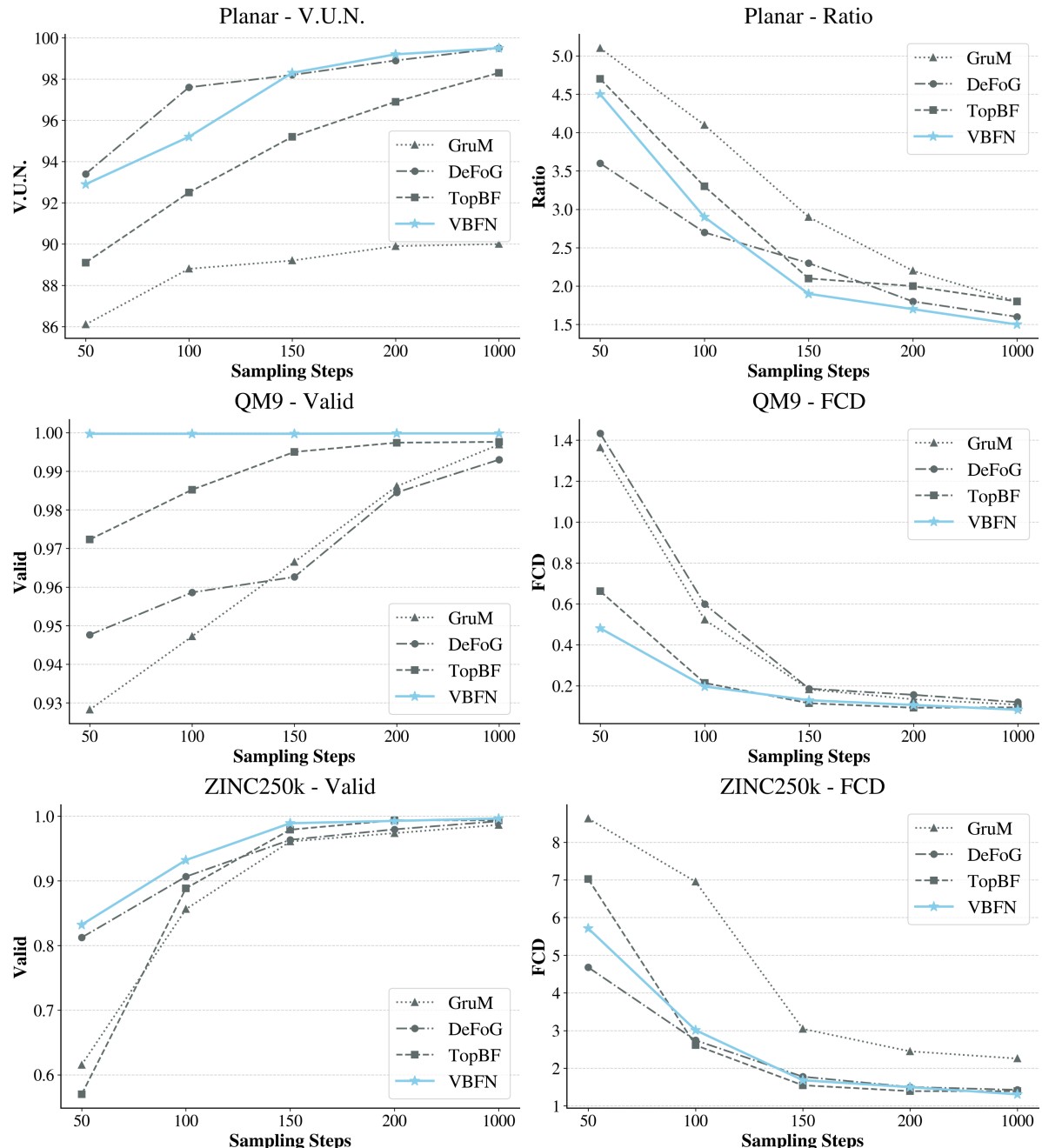

*Figure 2.* Metrics versus sampling steps on Planar, QM9, and ZINC250k. Higher is better for V.U.N./Valid; Lower is better for Ratio/FCD.

## C.3. Performance vs. Sampling Steps

We track how sample quality metrics evolve as the number of sampling steps increases to make a comparison among GruM, DeFoG, TopBF, and our VBFN. GruM is a strong Diffusion baseline on graph generation. DeFoG is the mightiest flow matching baseline with strong empirical performance. And TopBF represents a closely related BFN family that also relies on continuous parameterization and discretization at decoding.

Figure 2 illustrates how sample quality evolves as we increase the number of sampling steps. Overall, increasing steps consistently improves distributional metrics for all methods, but VBFN exhibits a more favorable tradeoff between quality and computation, as it reaches strong validity substantially earlier, while remaining competitive and often best on distributional

*Table 8.* Hyperparameters of VBFN.

| Category | Hyperparameter | Description | Value |
|---|---|---|---|
| Bayesian Flow | $\sigma_{1,\mathbf{X}}$ | Noise std for node channel | 0.2 |
| | $\sigma_{1,\mathbf{A}}$ | Noise std for edge channel | 0.2 |
| | $T$ | Sampling steps | $1,000$ |
| | $t_{\min}$ | Minimum time clamp | $10^{-4}$ |
| | `scale_X`, `scale_A` | Extra scaling (when enabled) | QM9/Synth: none; ZINC: 5, 1 |
| Precision Geometry | $\boldsymbol{\Omega}_{\text{prior}}^{X}$ mode | Node prior precision construction | `complete` |
| | $\boldsymbol{\Omega}_{\text{prior}}^{A}$ mode | Edge prior precision construction | `line_complete` |
| | $\boldsymbol{\Omega}_{\text{obs}}^{X}$ mode | Node observation precision mode | `diag_prior` |
| | $\boldsymbol{\Omega}_{\text{obs}}^{A}$ mode | Edge observation precision mode | `diag_prior` |
| | $\lambda_X$ | Coupling strength for node geometry | QM9: 1.0; ZINC/Synth: 0.2 |
| | $\lambda_A$ | Coupling strength for edge geometry | QM9: 1.0; ZINC/Synth: 0.2 |
| | $\varepsilon_X$ | Diagonal regularizer for nodes | QM9: $10^{-4}$; ZINC/Synth: $10^{-2}$ |
| | $\varepsilon_A$ | Diagonal regularizer for edges | QM9: $10^{-4}$; ZINC/Synth: $10^{-2}$ |
| Linear Solver | `solver` | SPD solver for the Bayesian update | cg |
| | `cg_max_iter` | Maximum CG iterations | 50 |
| | `cg_tol` | CG tolerance | $10^{-6}$ |
| | `cg_precond` | Preconditioner | `jacobi` |
| Optimization | `optimizer` | Optimizer | `AdamW` |
| | $\eta$ | Learning rate | $10^{-4}$ |
| | `wd` | Weight decay | $10^{-12}$ |
| | `bs` | Training batch size | QM9: $1,024$; ZINC250k: 256; Synth: 64 |
| | `epochs` | Training epochs | QM9/ZINC250k: $3,000$; Synth: $30,000$ |
| | `clip` | Gradient clipping norm | $10,000$ |
| Decoding / Masks | `eps_prob` | Numerical stabilization probability | $10^{-12}$ |
| | `mask_diag_edges` | Mask diagonal adjacency | `True` |

measures at larger step budgets.

On Planar, VBFN achieves the best V.U.N. among the compared methods throughout the practically relevant range by 200 steps, while also attaining the lowest Ratio across all step budgets, with a particularly pronounced gap at $1,000$ steps. This indicates that VBFN's coupled update improves structural consistency per step, reducing common Planar failure modes captured by Ratio. On QM9, VBFN attains near-saturated validity already at 50 steps and remains essentially perfect thereafter, whereas baselines require substantially more steps to approach the same validity. For FCD, methods are competitive at intermediate steps, but VBFN achieves the best final FCD at $1,000$ steps while retaining its early-step validity advantage. On ZINC250k, the benefit of VBFN is most evident in validity. VBFN is consistently the highest at almost every step count, demonstrating more information-efficient reverse updates on larger graphs. While DeFoG/TopBF can be competitive on FCD at some intermediate step budgets, VBFN catches up and achieves the best final FCD at $1,000$ steps, yielding the strongest overall tradeoff between validity and distributional matching.

In summary, these results support the interpretation that the structured, coupled Bayesian update makes each reverse step more effective. VBFN tends to reach usable validity in fewer steps and remains robust as the step budget increases, ultimately matching or surpassing the best distributional quality at longer trajectories.

### C.4. Class-order Permutation Robustness

Mapping classes to $[-1, 1]$ introduces an implicit ordering, which may induce an ordinal bias if the permutation of class indices changes the geometry. We added random class-order permutation tests on the molecular graph datasets. The synthetic datasets are not included because their node labels are binary (0 / 1), so there is no nontrivial class ordering to permute.

As shown in Table 7, the updated results show that the performance fluctuation is relatively small under different orderings on both QM9 and ZINC250k. The default order is not the sole source of gain. Some random orderings are even slightly better than the default order in certain indicators. Therefore, the ordering effect is at most a secondary inductive bias rather than the main contribution source.

## D. Experimental Details

### D.1. Hardware and Software Environment

To ensure fair and reproducible comparisons, all experiments were conducted under a fixed hardware and software environment. All models were trained and evaluated on a single NVIDIA A6000 GPU with an AMD EPYC 7763 (64 cores) @ 2.450GHz CPU. Experiments were implemented in PyTorch 2.0.1 with CUDA 11.7. Unless otherwise specified, we fix random seeds, use identical batch sizes and dataloader settings across methods, and average efficiency measurements over repeated runs.

### D.2. Backbone Network $\widehat{\boldsymbol{z}}_\phi$: Graph Transformer

VBFN instantiates $\widehat{\boldsymbol{z}}_\phi$ as a Graph Transformer, with the same architecture as the previous common practice (Vignac et al., 2023). The backbone takes the posterior belief $(\boldsymbol{\theta}_X, \boldsymbol{\theta}_A)$ and time $t$, together with padding masks, and outputs predictions that are used to form targets for both training and sampling.

### D.3. Datasets

**QM9 and ZINC250k.** QM9 and ZINC250k are standard molecular graph generation benchmarks. Nodes and edges are categorical (atom and bond types), represented by a continuous parameterization via class-centers (Appendix B.1) and decoded by CDF-based class probabilities. QM9 contains 133,885 molecules with up to 9 heavy atoms, while ZINC250k contains 249,455 molecules with up to 38 heavy atoms of 9 types, making ZINC250k substantially more demanding for scalability. Variable-size molecules are handled by padding and binary masks; adjacency is projected to satisfy symmetry and diagonal conventions.

**Planar and SBM.** Planar and SBM are synthetic graph benchmarks with 200 graphs, as well as $|N| = 64$ and $44 \leq |N| \leq 187$, respectively. Node attributes are continuous 2D coordinates $X \in \mathbb{R}^{N \times 2}$, while adjacency is binary. Accordingly, we regress $X$ with masked MSE and treat $A$ as categorical with CDF-based probabilities. We apply standard padding/masking and enforce symmetry in decoding.

**Tree.** Tree is a synthetic benchmark with 200 graphs and $|N| = 64$, whose node attributes are constant, while edges have multiple discrete types stored as a two-channel encoding $A \in \{0, 1\}^{N \times N \times 2}$. We treat the edge attribute as categorical and decode via CDF-based probabilities, with the same masking and symmetry projection as above.

### D.4. Hyperparameters

We summarize the key hyperparameters of VBFN in Table 8. Unless otherwise specified, we reuse the same configuration across datasets and only adjust dataset-dependent dimensions like numbers of classes or coordinate channels when necessary. This presentation is intended to make our experimental setup fully reproducible.

## E. Visualization

We visualize randomly selected samples generated by VBFN on all five datasets in Figures 3 and 4. These qualitative results complement the quantitative metrics reported in the main paper and appendix.

## F. Pseudocodes for VBFN

For completeness, we provide detailed training and sampling pseudocodes for VBFN in this section. The pseudocodes follow our implementation, including masking for variable-size graphs and the CDF-based decoding used for discrete attributes. They serve as a concise reference for reproducing both the coupled Bayesian update and the sampling procedure.

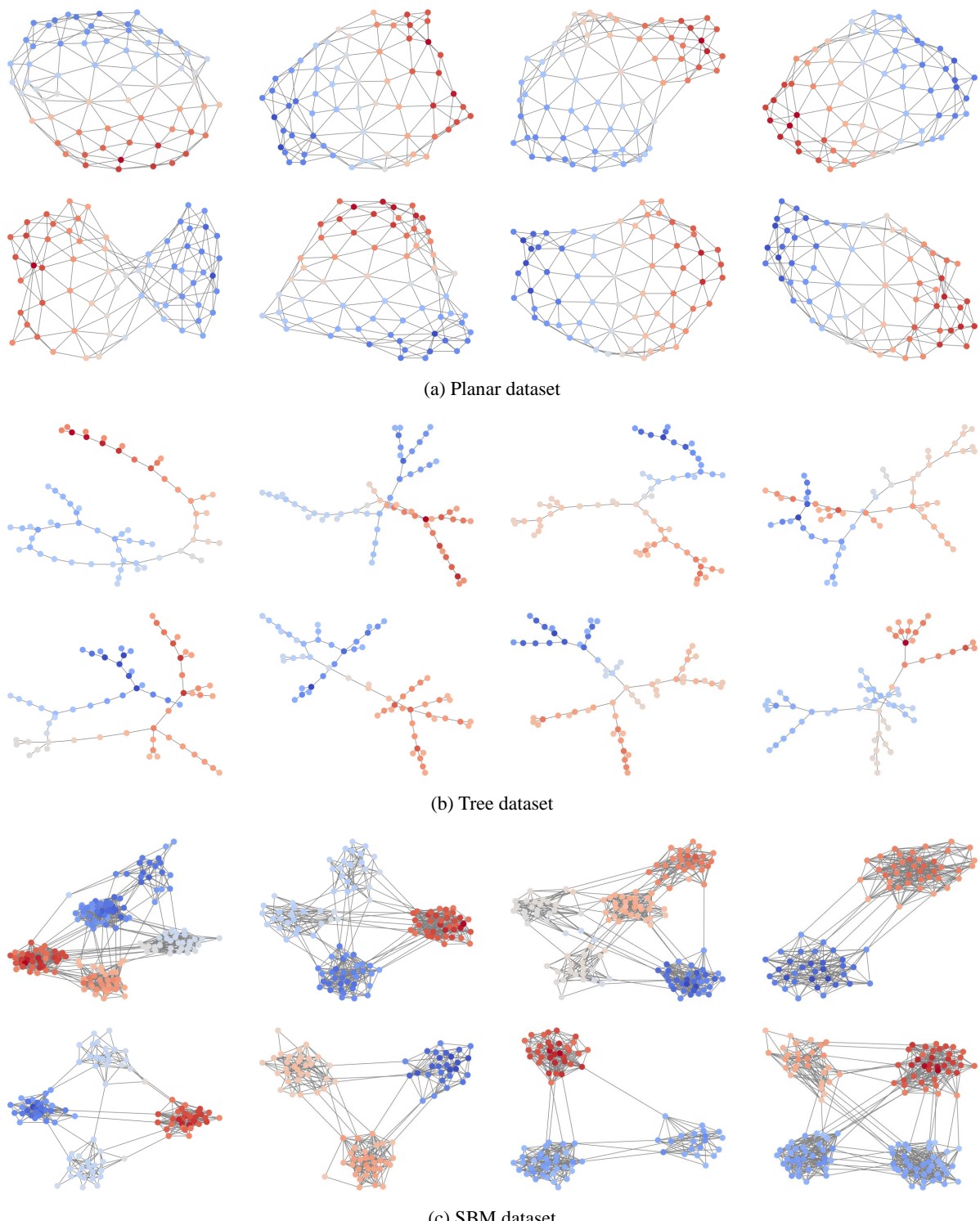

(a) Planar dataset

(b) Tree dataset

(c) SBM dataset

*Figure 3.* Visualization of generated graphs on synthetic graph datasets.

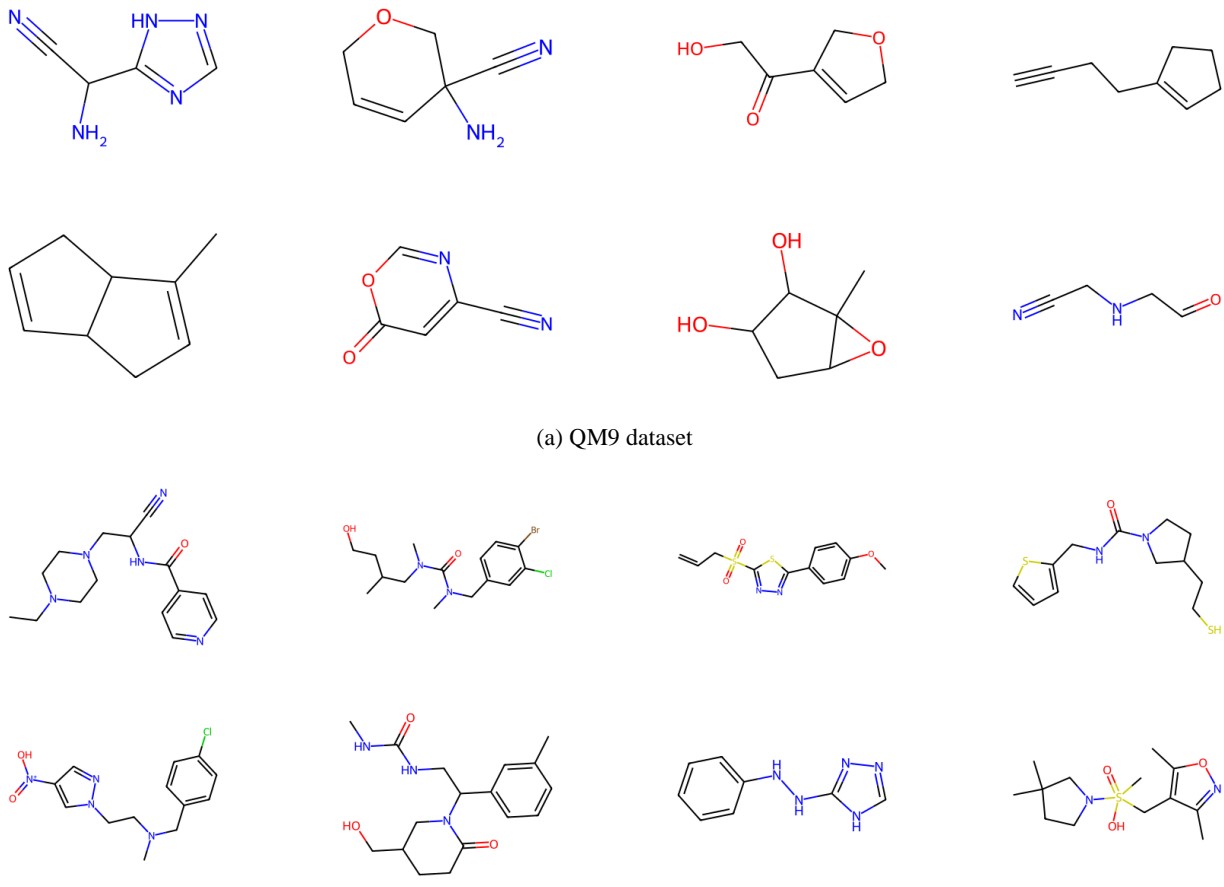

(a) QM9 dataset

(b) ZINC250k dataset

*Figure 4.* Visualization of generated graphs on molecular graph datasets.

---

**Algorithm 1** Training Algorithm

---

**Require:** Dataset of graphs $G = (X, A)$; schedules $\sigma_{1,X}, \sigma_{1,A} > 0$; $\varepsilon_X, \varepsilon_A > 0$; coupling strengths (encoded in $\mathcal{L}$).
**Require:** Discretization bins/intervals $\{[k_l^X, k_r^X]\}_{k=1}^{K_X}$, $\{[k_l^A, k_r^A]\}_{k=1}^{K_A}$ and centers $\{k_c^X\}$, $\{k_c^A\}$ (for discrete attributes).
    **Input:** a minibatch of graphs $\{G_b\}_{b=1}^B$ with node mask $\boldsymbol{m}^X \in \{0,1\}^{B \times N}$ and edge mask $\boldsymbol{m}^A \in \{0,1\}^{B \times N \times N}$.
    Sample $t \sim \mathcal{U}(0,1)$ and clamp $t \geq t_{\min}$.
    Compute $\beta_X \leftarrow \sigma_{1,X}^{-2t} - 1$,    $\beta_A \leftarrow \sigma_{1,A}^{-2t} - 1$.                                     (elementwise over batch)
    # Build structured precisions (nodes)
    Construct no-leakage dependency graph $\mathcal{H}_X$ and weighted Laplacian $\mathcal{L}_X$ from $(X, A)$ and mask $\boldsymbol{m}^X$.
    $\boldsymbol{\Omega}_{\mathrm{prior}}^X \leftarrow \boldsymbol{M}_X \mathcal{L}_X \boldsymbol{M}_X + \varepsilon_X \boldsymbol{I}$                                                     (Eq. (15))
    $\boldsymbol{\Omega}_{\mathrm{obs}}^X \leftarrow \textsc{BuildObs}(\boldsymbol{\Omega}_{\mathrm{prior}}^X, \boldsymbol{m}^X)$                                       (PRIOR / DIAG_PRIOR)
    # Build structured precisions
    Vectorize valid edges (e.g., upper-triangle) to get edge mask $\boldsymbol{m}^{A,\mathrm{vec}} \in \{0,1\}^{B \times E}$ and edge variable $A^{\mathrm{vec}}$.
    Construct line-graph dependency $\mathcal{H}_A$ and weighted Laplacian $\mathcal{L}_A$ on valid edges.
    $\boldsymbol{\Omega}_{\mathrm{prior}}^A \leftarrow \boldsymbol{M}_A \mathcal{L}_A \boldsymbol{M}_A + \varepsilon_A \boldsymbol{I}$.
    $\boldsymbol{\Omega}_{\mathrm{obs}}^A \leftarrow \textsc{BuildObs}(\boldsymbol{\Omega}_{\mathrm{prior}}^A, \boldsymbol{m}^{A,\mathrm{vec}})$                              (PRIOR / DIAG_PRIOR)
    # Sample flow state $\boldsymbol{\theta}_t$ in closed form
    **(nodes)** Sample $\boldsymbol{\epsilon}_X \sim \mathcal{N}(\boldsymbol{0}, \boldsymbol{I})$ and set
        $\boldsymbol{P}_X \leftarrow \boldsymbol{\Omega}_{\mathrm{prior}}^X + \beta_X \boldsymbol{\Omega}_{\mathrm{obs}}^X$,
        $\boldsymbol{r}_X \leftarrow \beta_X \boldsymbol{\Omega}_{\mathrm{obs}}^X X_c + \sqrt{\beta_X} \boldsymbol{\Omega}_{\mathrm{obs}}^{X\,1/2} \boldsymbol{\epsilon}_X$,
        $\boldsymbol{\theta}_X \leftarrow \textsc{SolveSPD}(\boldsymbol{P}_X, \boldsymbol{r}_X)$, then mask invalid nodes.
    **(edges)** Sample $\boldsymbol{\epsilon}_A \sim \mathcal{N}(\boldsymbol{0}, \boldsymbol{I})$ and analogously obtain $\boldsymbol{\theta}_A^{\mathrm{vec}}$, then unvectorize to $\boldsymbol{\theta}_A$ and apply edge mask.
    **if** $X$ is discrete **then**
        # Predict clean distribution parameters
        $(\boldsymbol{\mu}_X, \boldsymbol{\sigma}_X, \boldsymbol{\mu}_A, \boldsymbol{\sigma}_A) \leftarrow \widehat{\boldsymbol{z}}_\phi(\boldsymbol{\theta}_X, \boldsymbol{\theta}_A, t; \boldsymbol{m}^X, \boldsymbol{m}^A)$.
        # discretized probabilities via truncated Gaussian CDF on bins
        $p_X(k) \leftarrow \mathrm{CDF}(\boldsymbol{\mu}_X, \boldsymbol{\sigma}_X, k_r^X) - \mathrm{CDF}(\boldsymbol{\mu}_X, \boldsymbol{\sigma}_X, k_l^X)$                           $(k = 1..K_X)$
        $p_A(k) \leftarrow \mathrm{CDF}(\boldsymbol{\mu}_A, \boldsymbol{\sigma}_A, k_r^A) - \mathrm{CDF}(\boldsymbol{\mu}_A, \boldsymbol{\sigma}_A, k_l^A)$                           $(k = 1..K_A)$
        Apply masks and normalize $p_X, p_A$ over categories.
        $\widehat{X}_c \leftarrow \sum_k p_X(k) k_c^X$,      $\widehat{A}_c \leftarrow \sum_k p_A(k) k_c^A$.
    **else**
        # Continuous $X$ regression, discrete $A$ via CDF
        $(\widehat{X}_{\mathrm{reg}}, \boldsymbol{\mu}_A, \boldsymbol{\sigma}_A) \leftarrow \widehat{\boldsymbol{z}}_\phi(\boldsymbol{\theta}_X, \boldsymbol{\theta}_A, t; \boldsymbol{m}^X, \boldsymbol{m}^A)$.
        $p_A(k) \leftarrow \mathrm{CDF}(\boldsymbol{\mu}_A, \boldsymbol{\sigma}_A, k_r^A) - \mathrm{CDF}(\boldsymbol{\mu}_A, \boldsymbol{\sigma}_A, k_l^A)$                           $(k = 1..K_A)$
        Apply edge masks and normalize $p_A$ over categories.
        $\widehat{X}_c \leftarrow \widehat{X}_{\mathrm{reg}}$,      $\widehat{A}_c \leftarrow \sum_k p_A(k) k_c^A$.
    **end if**
    $\mathcal{L}_X \leftarrow \textsc{MaskedMSE}(\widehat{X}_c, X_c, \boldsymbol{m}^X)$,     $\mathcal{L}_A \leftarrow \textsc{MaskedMSE}(\widehat{A}_c, A_c, \boldsymbol{m}^A)$.
    # continuous-time weighting (matches $d\beta(t)$ discretization)
    $w_X(t) \leftarrow -\ln \sigma_{1,X} \cdot \sigma_{1,X}^{-2t}$,    $w_A(t) \leftarrow -\ln \sigma_{1,A} \cdot \sigma_{1,A}^{-2t}$.
    **return** $\mathcal{L}_\infty \leftarrow w_X(t)\mathcal{L}_X + w_A(t)\mathcal{L}_A$.

---

---

**Algorithm 2** Sampling Algorithm

---

**Require:** Number of samples $S$; steps $T$; schedules $\sigma_{1,X}, \sigma_{1,A}$; solver $\text{SOLVESPD} \in \{Cholesky, CG\}$.

  **Output:** generated graphs $\widehat{G} = (\widehat{X}, \widehat{A})$.

  **for** batch of size $B$ until $S$ samples are generated **do**

    Sample node counts and build node mask $\boldsymbol{m}^X$; derive edge mask $\boldsymbol{m}^A$ (optionally remove diagonal).

    Build $\boldsymbol{\Omega}_{\text{prior}}^X, \boldsymbol{\Omega}_{\text{obs}}^X$ and $\boldsymbol{\Omega}_{\text{prior}}^A, \boldsymbol{\Omega}_{\text{obs}}^A$ as in Algorithm 1 (sampling uses fixed prior mode; no data-dependent mode).

    # initialize natural parameters of the belief

    $\boldsymbol{P}_X \leftarrow \boldsymbol{\Omega}_{\text{prior}}^X, \quad \boldsymbol{h}_X \leftarrow \boldsymbol{\Omega}_{\text{prior}}^X \boldsymbol{\theta}_{0,X}$                  (typically $\boldsymbol{\theta}_{0,X} = \boldsymbol{0}$)

    $\boldsymbol{P}_A \leftarrow \boldsymbol{\Omega}_{\text{prior}}^A, \quad \boldsymbol{h}_A \leftarrow \boldsymbol{\Omega}_{\text{prior}}^A \boldsymbol{\theta}_{0,A}$.

    $\beta_X^{\text{prev}} \leftarrow 0, \quad \beta_A^{\text{prev}} \leftarrow 0$.

    **for** $i = 1$ **to** $T$ **do**

      $t \leftarrow \frac{i-1}{T}$, clamp $t \geq t_{\min}$.

      $\beta_X \leftarrow \sigma_{1,X}^{-2t} - 1, \quad \beta_A \leftarrow \sigma_{1,A}^{-2t} - 1$.

      $\alpha_X \leftarrow \max(\beta_X - \beta_X^{\text{prev}}, 0), \quad \alpha_A \leftarrow \max(\beta_A - \beta_A^{\text{prev}}, 0)$.

      $\beta_X^{\text{prev}} \leftarrow \beta_X, \quad \beta_A^{\text{prev}} \leftarrow \beta_A$.

      # current posterior means (coupled): $\boldsymbol{\theta} = \boldsymbol{P}^{-1}\boldsymbol{h}$

      $\boldsymbol{\theta}_X \leftarrow \text{SOLVESPD}(\boldsymbol{P}_X, \boldsymbol{h}_X)$, then apply node mask.

      $\boldsymbol{\theta}_A^{\text{vec}} \leftarrow \text{SOLVESPD}(\boldsymbol{P}_A, \boldsymbol{h}_A)$, then apply edge mask and unvectorize to $\boldsymbol{\theta}_A$.

      **if** $X$ is discrete **then**

        $(\boldsymbol{\mu}_X, \boldsymbol{\sigma}_X, \boldsymbol{\mu}_A, \boldsymbol{\sigma}_A) \leftarrow \widehat{\boldsymbol{z}}_\phi(\boldsymbol{\theta}_X, \boldsymbol{\theta}_A, t; \boldsymbol{m}^X, \boldsymbol{m}^A)$.

        $p_X(k) \leftarrow \text{CDF}(\boldsymbol{\mu}_X, \boldsymbol{\sigma}_X, k_r^X) - \text{CDF}(\boldsymbol{\mu}_X, \boldsymbol{\sigma}_X, k_l^X)$; normalize and mask.

        $p_A(k) \leftarrow \text{CDF}(\boldsymbol{\mu}_A, \boldsymbol{\sigma}_A, k_r^A) - \text{CDF}(\boldsymbol{\mu}_A, \boldsymbol{\sigma}_A, k_l^A)$; normalize and mask.

        Compute center means: $x_{\text{mean}} \leftarrow \sum_k p_X(k) k_c^X, \quad a_{\text{mean}} \leftarrow \sum_k p_A(k) k_c^A$; vectorize $a_{\text{mean}}$ to $a_{\text{mean}}^{\text{vec}}$.

        # sample incremental messages: $y \sim \mathcal{N}(\text{mean}, (\alpha \boldsymbol{\Omega}_{\text{obs}})^{-1})$

        Sample $\boldsymbol{\epsilon}_X, \boldsymbol{\epsilon}_A \sim \mathcal{N}(\boldsymbol{0}, \boldsymbol{I})$.

        $\boldsymbol{y}_X \leftarrow x_{\text{mean}} + (\alpha_X \boldsymbol{\Omega}_{\text{obs}}^X)^{-1/2} \boldsymbol{\epsilon}_X, \quad \boldsymbol{y}_A^{\text{vec}} \leftarrow a_{\text{mean}}^{\text{vec}} + (\alpha_A \boldsymbol{\Omega}_{\text{obs}}^A)^{-1/2} \boldsymbol{\epsilon}_A$.

      **else**

        $(\widehat{X}_{\text{reg}}, \boldsymbol{\mu}_A, \boldsymbol{\sigma}_A) \leftarrow \widehat{\boldsymbol{z}}_\phi(\boldsymbol{\theta}_X, \boldsymbol{\theta}_A, t; \boldsymbol{m}^X, \boldsymbol{m}^A)$.

        $\widehat{X}_c \leftarrow \widehat{X}_{\text{reg}}$.

        $p_A(k) \leftarrow \text{CDF}(\boldsymbol{\mu}_A, \boldsymbol{\sigma}_A, k_r^A) - \text{CDF}(\boldsymbol{\mu}_A, \boldsymbol{\sigma}_A, k_l^A)$; normalize and mask.

        Compute center means: $a_{\text{mean}} \leftarrow \sum_k p_A(k) k_c^A$; vectorize $a_{\text{mean}}$ to $a_{\text{mean}}^{\text{vec}}$.

        Sample $\boldsymbol{\epsilon}_X, \boldsymbol{\epsilon}_A \sim \mathcal{N}(\boldsymbol{0}, \boldsymbol{I})$.

        $\boldsymbol{y}_X \leftarrow \widehat{X}_c + (\alpha_X \boldsymbol{\Omega}_{\text{obs}}^X)^{-1/2} \boldsymbol{\epsilon}_X, \quad \boldsymbol{y}_A^{\text{vec}} \leftarrow a_{\text{mean}}^{\text{vec}} + (\alpha_A \boldsymbol{\Omega}_{\text{obs}}^A)^{-1/2} \boldsymbol{\epsilon}_A$.

      **end if**

      # Bayesian fusion in natural parameters

      $\boldsymbol{h}_X \leftarrow \boldsymbol{h}_X + \alpha_X \boldsymbol{\Omega}_{\text{obs}}^X \boldsymbol{y}_X, \quad \boldsymbol{P}_X \leftarrow \boldsymbol{P}_X + \alpha_X \boldsymbol{\Omega}_{\text{obs}}^X$.

      $\boldsymbol{h}_A \leftarrow \boldsymbol{h}_A + \alpha_A \boldsymbol{\Omega}_{\text{obs}}^A \boldsymbol{y}_A^{\text{vec}}, \quad \boldsymbol{P}_A \leftarrow \boldsymbol{P}_A + \alpha_A \boldsymbol{\Omega}_{\text{obs}}^A$.

    **end for**

    # final decode at $t = 1$

    $\boldsymbol{\theta}_X \leftarrow \text{SOLVESPD}(\boldsymbol{P}_X, \boldsymbol{h}_X), \quad \boldsymbol{\theta}_A \leftarrow \text{UNVEC}(\text{SOLVESPD}(\boldsymbol{P}_A, \boldsymbol{h}_A))$.

    **if** $X$ is discrete **then**

      $(\boldsymbol{\mu}_X, \boldsymbol{\sigma}_X, \boldsymbol{\mu}_A, \boldsymbol{\sigma}_A) \leftarrow \widehat{\boldsymbol{z}}_\phi(\boldsymbol{\theta}_X, \boldsymbol{\theta}_A, 1; \boldsymbol{m}^X, \boldsymbol{m}^A)$.

      $p_X(k) \leftarrow \text{CDF}(\boldsymbol{\mu}_X, \boldsymbol{\sigma}_X, k_r^X) - \text{CDF}(\boldsymbol{\mu}_X, \boldsymbol{\sigma}_X, k_l^X)$; normalize and mask.

      $p_A(k) \leftarrow \text{CDF}(\boldsymbol{\mu}_A, \boldsymbol{\sigma}_A, k_r^A) - \text{CDF}(\boldsymbol{\mu}_A, \boldsymbol{\sigma}_A, k_l^A)$; normalize and mask.

      $\widehat{X} \leftarrow \arg\max_k p_X(k), \quad \widehat{A} \leftarrow \arg\max_k p_A(k)$.

    **else**

      $(\widehat{X}_{\text{reg}}, \boldsymbol{\mu}_A, \boldsymbol{\sigma}_A) \leftarrow \widehat{\boldsymbol{z}}_\phi(\boldsymbol{\theta}_X, \boldsymbol{\theta}_A, 1; \boldsymbol{m}^X, \boldsymbol{m}^A)$.

      $p_A(k) \leftarrow \text{CDF}(\boldsymbol{\mu}_A, \boldsymbol{\sigma}_A, k_r^A) - \text{CDF}(\boldsymbol{\mu}_A, \boldsymbol{\sigma}_A, k_l^A)$; normalize and mask.

      $\widehat{X} \leftarrow \widehat{X}_{\text{reg}}, \quad \widehat{A} \leftarrow \arg\max_k p_A(k)$.

    **end if**

    Enforce masks and symmetry on $\widehat{A}$ (and convert to one-hot channels if required by the dataset).

  **end for**

---

# G. Future Work

A natural direction is to make the weighted Laplacian more expressive. In the current implementation, we fix the edge weights on the dependency graph. All incidence edges $(u, v) \in \mathcal{E}_{\text{inc}}$ share a global coefficient $\lambda_X$, and all symmetry edges $(u, v) \in \mathcal{E}_{\text{sym}}$ share a global coefficient $\lambda_A$. An appealing extension is to learn edge weights $\{w_{uv}\}_{(u,v)\in\mathcal{E}_{\mathcal{H}}}$ while keeping the dependency topology fixed, for instance, parameterizing $w_{uv}$ by a small network conditioned on local topology, or learning a structured reweighting with symmetry constraints. The weighted adjacency $W$ (and hence $\mathcal{L} = \Delta - W$) would then become data-adaptive, while the prior precision remains SPD via $\Omega_{\text{prior}} = M\mathcal{L}M + \varepsilon I$.

