# OpenReview forum: "Variational Bayesian Flow Network for Graph Generation"
_ICML.cc/2026/Conference — ICML 2026 regular_

### Official Review · Reviewer_xMWH · 2026-03-10

**Soundness:** 3
**Presentation:** 3
**Significance:** 2
**Originality:** 3
**Overall Recommendation:** 4
**Confidence:** 3

**Summary:**

This paper introduces node–edge coupling within the Bayesian Flow Network (BFN) framework, allowing structural information to propagate during the Bayesian update process.

**Compliance With Llm Reviewing Policy:**

Affirmed.

**Final Justification:**

I find this paper solid and complete in terms of method and presentation.

My concerns are addressed.

I keep my positive rating on it.

**Key Questions For Authors:**

1. The coupling structure is manually designed based on node-edge relationships. Have the authors explored learning this structure or evaluating the sensitivity to different coupling designs except the ones provided in Table 3?

2. Independent noise modeling works well in other domains such as images and text. Could the authors elaborate more clearly on why explicit node-edge coupling is particularly important for graph generation?

3. Could a similar node–edge coupling also be implemented within diffusion or discrete flow matching (DFM) frameworks? If so, what would be the main differences in implementation? Why is it necessary to introduce this mechanism specifically within the BFN framework?

**Limitations:**

I did not find clear discussion about the limitations of this work.
A possible limitation is that this framework may be specific to BFN, rather than being easily extended to more general classes of models, which limits largely its generality. The scalability to larger graphs is not sufficiently clear from the current paper.

**Strengths And Weaknesses:**

Strengths

1. The paper is generally well written and the method is clearly motivated from the perspective of structural coupling in graph generation.

2. The idea of introducing a structured precision matrix into the BFN framework is conceptually simple and provides a clean probabilistic interpretation.

3. Empirical results show improvements on several structural metrics (e.g., FCD and scaffold similarity) on molecular datasets.

4. The ablation study demonstrates that the proposed node-edge coupling plays an important role in the model performance.

Weaknesses

1. The proposed approach introduces additional computational overhead, but the paper provides the training or sampling efficiency analysis in the Appendix. This analysis can be put earlier.

2. The dependency graph and coupling structure are hand-designed rather than learned, and it is unclear how sensitive the performance is to these design choices.

---

> ### Author Rebuttal · Authors · 2026-03-31
>
> # Response to Reviewer xMWH
>
> We thank the reviewer for the constructive feedback. We address the main points below.
>
> Supplementary Experiments: https://anonymous.4open.science/r/VBFN-D6B0/Supplementary_Experiments.md
>
> > **Concern 1**: Computation, scalability, and placement of efficiency analysis
>
> ***Response***. Thanks for this critical point. We agree the efficiency discussion should appear earlier and that the current limitations should be stated more directly in the main paper. Our claim is not that VBFN is uniformly better in every resource dimension, but that coupled Bayesian updates remain feasible on molecular graph benchmarks with a clear runtime-memory trade-off. To make this explicit, we added a stage-wise memory profile and an additional wall-clock/memory comparison. Supplementary Tables 3 and 4 show that the main persistent sampling cost comes from dense edge-space precisions and posterior caches rather than from the CG iterations themselves, while VBFN(CG) remains competitive in wall-clock time on QM9 and ZINC250k but uses noticeably more sampling memory on ZINC250k (presented in the anonymous link). We will move this discussion forward from the appendix, state large-graph scalability more clearly as a current limitation, and summarize the trade-off in the Conclusion.
>
> > **Concern 2**: On hand-designed dependency/coupling structure, sensitivity, and learnable extensions
>
> ***Response***. Thanks for this important point. We do not claim that the current hand-designed coupling is universally optimal, nor that learned coupling is unhelpful. Our goal is narrower: to isolate whether putting node-edge coupling directly into the Bayesian update geometry is beneficial. We therefore use a minimal, sample-agnostic, and interpretable coupling induced by incidence/symmetry relations, instead of simultaneously introducing a second trainable structure-learning problem. If the coupling itself were learned, the gains would be confounded by additional parameterization, regularization, and SPD-stability issues on the precision operator, making it harder to attribute improvements specifically to the structured Bayesian geometry.
>
> To address sensitivity directly, we added two controlled ZINC250k studies. Supplementary Table 5 shows that a local grid search of $(\lambda_X,\lambda_A)$ around the default setting yields a reasonably stable neighborhood rather than a brittle single point, with edge-side coupling somewhat more sensitive than node-side coupling. Supplementary Table 6 perturbs the coupling template under a matched coupling budget; the designed incidence/symmetry template consistently outperforms rewired and random alternatives, indicating that the gain comes from graph-aligned node-edge relations rather than merely adding more edges to the dependency graph. We will clarify in the revision that learned coupling is an important future direction, but a different question from the one isolated here.
>
> > **Concerns 3 \& 4**: Why is explicit node-edge coupling important for graph generation, and why is BFN a particularly natural framework for it?
>
> ***Response***. Thanks for raising these important questions. Independent-noise modeling can work well in images and text, but graph generation differs in one key respect: graph quality depends not only on whether node and edge variables are individually plausible, but also on whether they are jointly consistent. In molecular graphs, plausible atom types and bond patterns are not sufficient when considered separately; distributional fidelity depends on their joint configuration. Under a factorized Bayesian update, this consistency is not represented in the update geometry itself and must be recovered entirely by the backbone after the fact.
>
> This is where VBFN differs. Its posterior precision
> $$
> P_t=\Omega_{\mathrm{prior}}+\beta(t)\Omega_{\mathrm{obs}}
> $$
> allows structural information to propagate during the update, so evidence about one part of the graph can immediately reshape beliefs about coupled parts. Our ablations support this point: no coupling or one-sided coupling leads to clear failure modes, while full coupling is required to simultaneously maintain high validity and strong structural metrics.
>
> We do not claim such coupling is exclusive to BFN. Similar ideas can in principle be implemented in diffusion or DFM, where structure is typically injected into the denoiser, score, velocity, transition design, or an auxiliary guidance term. Our claim is narrower: BFN provides a particularly natural probabilistic route to make node-edge coupling intrinsic to the Bayesian update itself, rather than adding it only as an external regularizer or architectural bias. This is also why the current formulation is both a strength and a boundary of the work: the idea is not conceptually exclusive to BFN, but our operator-based formulation is most cleanly realized within BFN. We will make this scope explicit in the revision and summarize it in the limitations discussion.

---

> > ### Author Rebuttal · Reviewer_xMWH · 2026-04-02
> >
> > My concerns have been adequately addressed. I keep the current positive feedback to this submission.

---

> > > ### Author Response · Authors · 2026-04-04
> > >
> > > Thank you very much for your positive feedback and for taking the time to revisit our response. We really appreciate your effort, and we are glad that our response has adequately addressed your concerns.
> > >
> > > If you feel a score raise is appropriate, we would be very grateful.

---

### Official Review · Reviewer_Jj4A · 2026-03-11

**Soundness:** 3
**Presentation:** 2
**Significance:** 3
**Originality:** 3
**Overall Recommendation:** 4
**Confidence:** 3

**Summary:**

This paper proposes VBFN, a graph generation method that replaces the factorized belief geometry of classical Bayesian Flow Networks with a joint Gaussian belief defined by structured precision matrices. The main idea is to explicitly couple node and edge variables during Bayesian updates, so inference becomes a structured linear solve rather than an element-wise update.

**Compliance With Llm Reviewing Policy:**

Affirmed.

**Final Justification:**

The paper presents an interesting and conceptually clean approach to graph generation by replacing the factorized belief geometry of classical Bayesian Flow Networks with a structured joint Gaussian belief over coupled node and edge variables. I continue to view the main strengths as the clear motivation, the operator-level formulation of the posterior update, and the fact that the method retains a relatively tractable analytical form despite introducing structured coupling.

My original concerns were mainly about three points: the meaning of the claim about “avoiding leakage,” the statistical justification for the specific hand-designed coupling structure and prior weights, and the extent to which the framework depends on the tied-covariance assumption. The rebuttal addresses these concerns to a meaningful extent. In particular, it sharpens the leakage claim to one about the absence of direct sample-specific leakage rather than the absence of structural inductive bias, clarifies that the proposed prior is intended as a minimal and interpretable sample-agnostic structured prior rather than a universally optimal one, and explains more clearly that tied covariance is a deliberate BFN-consistent design choice for tractability rather than a claim that more general covariance choices are impossible.

That said, some concerns remain only partially resolved. I still think the statistical case for why this particular structured prior should be preferred over other plausible structured priors is not fully established, and the generality of the theory beyond the tied-covariance setting remains somewhat limited. These limitations make the scope of the contribution narrower than the broadest framing might initially suggest. However, the rebuttal is substantive, technically responsive, and makes the paper better scoped and easier to evaluate.

Overall, the rebuttal reinforces rather than overturns my prior assessment. I find the paper to have real originality and reasonable technical merit, with weaknesses that are important but sufficiently understood. I therefore keep my score at 3.

**Key Questions For Authors:**

See "Weaknesses".

**Limitations:**

See "Weaknesses".

**Strengths And Weaknesses:**

**Strengths**
1. The paper has a clear and meaningful motivation: graph generation naturally involves coupled node-edge structure, and the method tries to encode that coupling directly in the Bayesian update itself rather than leaving it entirely to the backbone network.
2. The method is not only conceptually novel, but also retains a relatively clean analytical form: the posterior update is closed-form in operator form, and the training objective remains tractable under the tied-covariance design.

**Weaknesses**
1. The paper argues that the dependency graph avoids leakage because it depends only on representation structure and not on sample-specific adjacency values. However, this is mostly a construction-level argument. The paper does not give a more formal notion of leakage, nor does it analyze whether the chosen structural prior could still introduce strong implicit biases correlated with the unknown graph structure.
2. The prior is built from incidence and symmetry couplings with manually specified weights such as $\lambda_X$ and $\lambda_A$, and the paper mainly shows that this construction is valid and solvable. What is missing is a stronger justification for why this particular coupling structure is the right one statistically, or under what conditions it is preferable to other possible structured priors.
3. The structured KL objective becomes simple largely because the sender and receiver use tied channel covariance, which cancels the extra Gaussian KL terms. This keeps the method analytically clean, but it also makes the framework rely on a fairly restrictive assumption, so it is less clear how well the theory extends to more general covariance choices.

---

> ### Author Rebuttal · Authors · 2026-03-31
>
> # **Response** to Reviewer Jj4A
>
> We thank the reviewer for the careful theoretical reading. We address the concerns by distinguishing more clearly between what is formally claimed, what is used as inductive bias, and what is deliberately chosen for tractability. For completeness, the latest supplementary experiments are available at the anonymous link below.
>
> Supplementary Experiments: https://anonymous.4open.science/r/VBFN-D6B0/Supplementary_Experiments.md
>
> ------
>
> > **Concern 1**: On "avoiding leakage'' vs. "introducing structural bias"
>
> ***Response***. We thank you for this important point. Our original wording was too construction-level, and we will sharpen the claim to **no direct sample-specific leakage**. Let $\mathcal R$ denote the representation metadata (tensor indexing, masks, symmetry convention, parameterization type), excluding sample-specific adjacency values. In our construction, $\mathcal H$ and $\Omega_{\mathrm{prior}}$ are deterministic functions of $\mathcal R$ only. Equivalently, $(\mathcal H,\Omega_{\mathrm{prior}})\perp A \mid \mathcal R$.
>
> This does not imply absence of bias. Rather, the prior introduces a deliberate **sample-agnostic** structural inductive bias at the representation level. Such relational inductive bias is standard and widely accepted in graph learning [1]. Our claim is therefore only the absence of **direct** leakage from the current sample’s unknown adjacency, not the absence of inductive bias. We will revise the wording accordingly.
>
>
>
> > **Concern 2**: On hand-designed coupling structure and $\lambda_X,\lambda_A$
>
> ***Response***. Thanks for raising this important point. We would like to clarify that our claim is narrower than *this is the statistically right structured prior in general*. Rather, it is a minimal, sample-agnostic, and interpretable GMRF prior aligned with the graph representation. Its key statistical meaning is the Laplacian Dirichlet form already defined in Eq. (16),
> $$
> \mathcal{J}(\mathbf s)\triangleq \mathbf s^\top \mathbf{\mathcal L}\mathbf s=\frac{1}{2}\sum_{(u,v)\in\mathcal{E_H}} w_{uv}(s_u-s_v)^{2},
> $$
> which penalizes disagreement only along adjacent pairs in the representation-level dependency graph $\mathcal H$, rather than imposing arbitrary global correlations. In this sense, $\lambda_X$ and $\lambda_A$ control the strength of this structural bias through the edge weights $\{w_{uv}\}$. We therefore view this prior as preferable when graph quality depends on coherent node-edge propagation under sample-agnostic prior information, without claiming that it dominates richer learned or higher-order priors in general. This interpretation is also consistent with our current ablations: disabling coupling or keeping only one side causes clear failure modes, and a local $\lambda$-sensitivity check (see Supplementary Table 5 in the anonymous link) shows that the reported setting is the best among the tested local combinations. We will clarify this scope and the role of $\lambda_X,\lambda_A$ in Sec. 4.2.
>
> > **Concern 3**: On the tied-covariance assumption
>
> ***Response***. Thank you for this important point. We would like to clarify that tied covariance is not introduced in VBFN merely to simplify the KL. Instead, it is inherited from the original Gaussian BFN message-matching construction. In classical BFNs, the receiver is defined by
> $$
> p_R(y \mid \theta_t;t)=\mathbb E_{p_O(z \mid \theta_t;t)}[p_S(y \mid z;t)].
> $$
> Hence, if the model outputs a point estimate $p_O(z \mid \theta_t;t)=\delta(z-\hat z_\phi(\theta_t,t))$, then $p_R(y \mid \theta_t;t)=p_S(y \mid \hat z_\phi(\theta_t,t);t)$, so the receiver must inherit the sender covariance [2]. Otherwise, $p_R$ would no longer be the receiver induced by the BFN definition above, and the KL would mix mean-prediction error with channel-mismatch error rather than only reflecting message mismatch.
>
> VBFN keeps exactly this principle, while lifting the isotropic covariance to the structured one $(\beta(t)\Omega_{\mathrm{obs}})^{-1}$. The tied choice affects the training-side simplification in Proposition 4.4,
> $$
> D_{\mathrm{KL}}(p_S || p_R)=\frac{\beta(t)}{2}||z-\hat z_\phi(\theta_t,t)||_{\Omega}^2.
> $$
>
>
> whereas the coupled Bayesian update in Theorem 4.1,
> $$
> P_t\triangleq\Omega_{\mathrm{prior}}+\beta(t)\Omega_{\mathrm{obs}},
> $$
> does not rely on this KL cancellation. More general covariance choices are possible if the model outputs a non-degenerate covariance, but then the extra Gaussian covariance terms remain and Proposition 4.4 no longer applies. We will revise the text to make this scope explicit: tied covariance is a deliberate, BFN-consistent, and tractable design choice, not a claim that more general covariance choices are impossible.
>
> [1] Battaglia P W, Hamrick J B, Bapst V, et al. Relational inductive biases, deep learning, and graph networks[J]. arXiv preprint arXiv:1806.01261, 2018.
>
> [2] Graves A, Srivastava R K, Atkinson T, et al. Bayesian flow networks[J]. arXiv preprint arXiv:2308.07037, 2023.

---

> > ### Author Rebuttal · Reviewer_Jj4A · 2026-04-05
> >
> > The rebuttal is thoughtful and addresses my concerns to a meaningful extent. In particular, it sharpens the scope of the claims, clarifies which parts are intended as formal guarantees versus deliberate modeling choices for tractability, and provides a more careful explanation of the role of structural bias in the proposed prior. I also appreciate the authors’ effort to distinguish the absence of direct sample-specific leakage from the presence of sample-agnostic inductive bias, which makes the central claim more precise and more credible.
> >
> > That said, some concerns remain only partially resolved. Most notably, I still think the statistical justification for the specific structured prior is not fully established, and the extent to which the framework generalizes beyond the tied-covariance setting remains somewhat limited. These issues do not invalidate the paper, but they do leave some open questions about the full scope of the theoretical contribution.
> >
> > Overall, the response improves my confidence in the paper and makes the contribution clearer and better scoped. While a few of my original concerns remain only partially addressed, the rebuttal is substantive and helpful, and I will keep my score unchanged.

---

> > > ### Author Response · Authors · 2026-04-06
> > >
> > > We thank the reviewer for the thoughtful follow-up and for acknowledging our response clarified the contribution and sharpened the scope of our claims. Below we address the remaining points.
> > >
> > > > **Concern**: On the hand-designed coupling structure and whether the proposed prior is sufficiently justified statistically.
> > >
> > > ***Response***. We would like to clarify that **universal optimality over all structured priors is not the central proposition of this paper**, and is not well-posed without specifying a prior class, a data-generating family, and a comparison criterion. Our claim is that VBFN introduces a **sample-agnostic, representation-aligned, and tractable structural geometry** directly into the Bayesian update.
> > >
> > > The relevant question is therefore whether the prior is principled for the claimed mechanism. We believe it is. We construct a dependency graph $\mathcal H$ from incidence and symmetry relations already implied by the representation, and define
> > > $$
> > > \Omega_{\mathrm{prior}}=M\mathcal LM+\varepsilon I.
> > > $$
> > > This yields a sample-agnostic, sparse, interpretable, and SPD precision. Its statistical meaning is explicit through
> > > $$
> > > \mathcal J(\mathbf s)=\mathbf s^\top\mathcal L\mathbf s=\frac12\sum_{(u,v)\in\mathcal E_{\mathcal H}}w_{uv}(s_u-s_v)^2,
> > > $$
> > > which penalizes disagreement only along representation-coupled pairs, not arbitrary global correlations. Thus $\lambda_X,\lambda_A$ control two specific coupling mechanisms determined by the representation, rather than an unconstrained prior family.
> > >
> > > Accordingly, “why this prior rather than every other structured prior” is not the decisive standard here. Once one requires the prior to be sample-agnostic, representation-consistent, sparse, and compatible with closed-form coupled inference, the Laplacian-GMRF construction is a natural minimal choice. It directly yields
> > > $$
> > > P_t=\Omega_{\mathrm{prior}}+\beta(t)\Omega_{\mathrm{obs}},\qquad
> > > P_t\theta_t=\Omega_{\mathrm{prior}}\theta_0+\beta(t)\Omega_{\mathrm{obs}}y,
> > > $$
> > > so the prior is not an auxiliary regularizer, but the geometric object defining the coupled inference itself.
> > >
> > > Our experiments are targeted to this proposition. The ablations show that removing coupling, keeping only one side, or perturbing designed template causes systematic failure modes, while the local $\lambda$-sensitivity study supports the default setting within the same template. We will revise the paper to make this scope explicit: we do **not** claim full-general optimality, but a principled minimal prior for the claimed setting, for which current theory and ablations are sufficient.
> > >
> > > > **Concern**: On the tied-covariance assumption and whether this is an overly restrictive design choice.
> > >
> > > ***Response***. We would like to clarify that the tied-covariance choice in VBFN is **not** introduced merely for convenience. It follows directly from the original Gaussian BFN construction [1]. More general covariance choices are possible, but they correspond to a different objective and are outside the present formulation.
> > >
> > > In the original BFN framework, the receiver is defined as
> > > $$
> > > p_R(y\mid \theta;t,\alpha)=\mathbb E_{p_O(x'\mid\theta,t)}[p_S(y\mid x';\alpha)].
> > > $$
> > > For the Gaussian case,
> > > $$
> > > p_S(y\mid x;\alpha)=\mathcal N(y;x,\alpha^{-1}I),\qquad
> > > p_O(x\mid\theta,t)=\delta(x-\hat x(\theta,t)),
> > > $$
> > > which gives
> > > $$
> > > p_R(y\mid\theta;t,\alpha)=\mathcal N(y;\hat x(\theta,t),\alpha^{-1}I).
> > > $$
> > > Hence the receiver necessarily inherits the sender covariance. What changes is the mean, **not** the channel covariance. Because the covariance is shared, the KL reduces to
> > > $$
> > > D_{\mathrm{KL}}(p_S||p_R)=D_{\mathrm{KL}}\left(\mathcal N(x,\alpha^{-1}I)||\mathcal N(\hat x(\theta,t),\alpha^{-1}I)\right)=\frac{\alpha}{2}||x-\hat x(\theta,t)||_2^2,
> > > $$
> > > which is precisely the clean Gaussian BFN objective.
> > >
> > > VBFN preserves this principle, replacing the isotropic covariance by $(\beta(t)\Omega_{\mathrm{obs}})^{-1}$:
> > > $$
> > > p_S(y\mid z;t)=\mathcal N\left(y;z,(\beta(t)\Omega_{\mathrm{obs}})^{-1}\right),\qquad
> > > p_R(y\mid\theta_t;t)=\mathcal N\left(y;\hat z_\phi(\theta_t,t),(\beta(t)\Omega_{\mathrm{obs}})^{-1}\right).
> > > $$
> > > This is the structured analogue of the original Gaussian BFN construction.
> > >
> > > If one instead allows an **untied** receiver covariance $\Sigma_R\neq(\beta(t)\Omega_{\mathrm{obs}})^{-1}$, the receiver is no longer the standard BFN receiver induced by the sender under a point-estimate output. The KL then contains additional covariance terms, so the objective no longer isolates mean-level message mismatch and instead entangles prediction error with channel mismatch. This changes the training semantics and removes the clean quadratic objective of Proposition 4.4.
> > >
> > > We will revise the paper to state this scope explicitly. Tied covariance is used not only for tractability, but because it preserves the original Gaussian BFN message-matching semantics in the structured setting.
> > >
> > > [1] Graves A, Srivastava R K, Atkinson T, et al. Bayesian flow networks[J]. arXiv preprint arXiv:2308.07037, 2023.

---

### Official Review · Reviewer_7HL7 · 2026-03-12

**Soundness:** 3
**Presentation:** 3
**Significance:** 3
**Originality:** 3
**Overall Recommendation:** 4
**Confidence:** 3

**Summary:**

The manuscript studies a meaningful question: whether graph generation benefits from replacing the factorized belief geometry of classical Bayesian Flow Networks with a coupled variational belief defined by structured precision matrices. More concretely, the paper asks whether node-edge dependence should be built into the Bayesian fusion mechanism itself rather than left entirely to the backbone network. To this end, the paper proposes VBFN, where the posterior update is carried out under a joint Gaussian belief and reduces to solving a symmetric positive definite linear system. The empirical results on synthetic graph benchmarks and molecular generation benchmarks are strong overall. My overall view is positive, but I think the current evidence supports a somewhat narrower claim than the broadest framing in the paper.

**Compliance With Llm Reviewing Policy:**

Affirmed.

**Key Questions For Authors:**

**1. Observation precision and the central claim.**
The default and best-performing configuration uses a diagonalized observation precision. How should the reader interpret the central claim in light of this? Does the paper mainly support the value of a structured prior geometry, or does it also support the practical value of a fully structured observation channel?

**2. Robustness to class-order permutations.**
Have the authors tested robustness to permutations of the discrete class order? This would help separate the benefit of the proposed Bayesian coupling from the effect of the induced continuous class geometry.

**3. Sampling-memory overhead.**
The sampling-memory overhead on ZINC250k is still large. Can the authors clarify what dominates this cost in practice?

**4. No backpropagation through the linear solver.**
The paper states that it does not backpropagate through the linear solver. Should this be viewed mainly as an implementation choice for efficiency and memory, or as part of the intended training formulation?

**5. Choice of efficiency reference.**
Since DeFoG and TopBF are also important baselines in the empirical section, it would help the presentation to briefly explain in the main paper why GruM is used as the main efficiency reference.

**Limitations:**

Partially. The paper acknowledges limitations, but I think the current discussion should be more direct. In particular, the possible ordering effect from the continuous class parameterization and the sampling-memory cost on larger graphs should be stated more clearly.

**Strengths And Weaknesses:**

**Strengths**

**1. Meaningful modeling question.**
The paper studies a meaningful modeling question. For graph generation, it is reasonable to ask whether coupling should appear in the generative geometry itself.

**2. Technically coherent method.**
The dependency graph, the structured prior precision, the observation precision, and the coupled Bayesian update fit together cleanly.

**3. Strong empirical results overall.**
The method performs very well on both synthetic and molecular benchmarks, and the coupling ablations are informative.

**4. Useful computational evaluation.**
The paper makes a useful effort to evaluate computation, including solver choice and separate reporting of training and sampling cost.

**Weaknesses**

**1. The main claim is broader than the current evidence.**
The paper presents VBFN as a structured coupled Bayesian fusion mechanism, but the strongest practical configuration uses a diagonalized observation precision rather than the more direct structured choice $\Omega_{\mathrm{obs}} = \Omega_{\mathrm{prior}}$. As a result, the paper more clearly shows that a structured prior geometry is useful than that a fully structured observation channel is practically effective.

**2. The discrete parameterization is not fully neutral.**
Classes are mapped to continuous centers and decoded by CDF-based probabilities, which introduces an implicit ordering. Part of the gain may therefore depend on the induced class geometry rather than only on the proposed coupled Bayesian mechanism. Robustness to class-order permutations would help separate these effects.

**3. The scalability story is mixed.**
The efficiency appendix is useful, but the sampling memory on ZINC250k remains substantially higher than the GruM baseline even when CG is used.

**4. Some method-critical caveats are not discussed prominently enough in the main paper.**
In particular, the observation-precision tradeoff, the possible class-ordering effect from the continuous class parameterization, and the memory overhead are central to understanding the current boundary of the method.

---

> ### Author Rebuttal · Authors · 2026-03-31
>
> # **Response** to Reviewer 7HL7
>
> We sincerely thank for your constructive feedback and valuable suggestions. We have carefully addressed all concerns below.
>
> Supplementary Experiments: https://anonymous.4open.science/r/VBFN-D6B0/Supplementary_Experiments.md
>
> ------
>
> > **Concern 1**: On the central claim: structured prior geometry vs. fully structured observation channel
>
> ***Response***. Thanks for raising this critical point. We would like to clarify that the theory only requires a fixed sample-agnostic $\Omega_{\text{obs}}\succ 0$. It does **not** require $\Omega_{\text{obs}}$ to be fully structured. In fact, our ablations explain why the best practical configuration uses a diagonalized observation precision.
>
> If one directly sets $\Omega_{\text{obs}}=\Omega_{\text{prior}}$, then
> $$
> P_t \triangleq \Omega_{\text{prior}}+\beta(t)\Omega_{\text{obs}}=(1+\beta(t))\Omega_{\text{prior}},
> $$
> and therefore
> $$
> \theta_t = P_t^{-1}\left(\Omega_{\text{prior}}\theta_0+\beta(t)\Omega_{\text{prior}}y\right)=\frac{\theta_0+\beta(t)y}{1+\beta(t)}.
> $$
> Thus, the structured precision **cancels** in the update, and the posterior collapses to a simple convex combination of $\theta_0$ and $y$, weakening prior-induced propagation. A diagonalized observation channel avoids this collapse, injects evidence in a more stable coordinate-local way, and still preserves coupling through $\Omega_{\text{prior}}$.
>
> So our refined claim is: the paper’s strongest empirical evidence supports **structured prior + coupled Bayesian update**, rather than *fully structured observation channel is always empirically best*.
>
> ---
>
> > **Concern 2**: On class-order permutation robustness
>
> ***Response***. Thanks for this important question. We agree that mapping classes to continuous centers in $[-1,1]$ may introduce an implicit ordering effect, so we added random class-order permutation tests on the molecular graph datasets. The synthetic datasets are not included because their node labels are binary (0/1), so there is no nontrivial class ordering to permute.
>
> As shown in Supplementary Table 2 (presented in the the anonymous link), the updated results show that the performance fluctuation is relatively small under different orderings on both QM9 and ZINC250k. The default order is not the sole source of gain. Some random orderings are even slightly better than the default order in certain indicators. Therefore, the ordering effect is at most a secondary inductive bias rather than the main contribution source.
>
> ------
>
> > **Concern 3**: On scalability and memory and Supplementary Choice of efficiency reference
>
> ***Response***. Thanks for this important question. To clarify what dominates the sampling memory cost in practice, we added a stage-wise memory profile for ZINC250k sampling with the CG solver. As shown in Supplementary Table 3, node-space costs are negligible, whereas constructing the dense edge-space precisions $\Omega_{\mathrm{prior}}^{A}$ and $\Omega_{\mathrm{obs}}^{A}$ already raises allocated memory from 0.07 GB to 9.28 GB, and initializing the edge posterior cache $P_A$ further raises it to 13.91 GB. By contrast, the CG solve itself peaks at only 14.03 GB. Thus, the main persistent bottleneck comes from the dense edge-space coupled geometry and cached posterior states, while the largest transient peaks arise in prediction/decoding and Bayes fusion. We will clarify this point in the revision.
>
> Moreover, we also added wall-clock time and memory comparisons against GruM, DeFoG, and TopBF on QM9 and ZINC250k. As shown in Supplementary Table 4, these additional results show that VBFN(CG) remains competitive in training and sampling time, especially on ZINC250k, while its sampling memory is noticeably higher. We therefore agree that practical scalability should be discussed as a runtime-memory trade-off rather than a uniformly stronger efficiency claim. We will revise the paper accordingly and move this discussion forward from the appendix.
>
> ------
>
> > **Concern 4**: No backpropagation through the linear solver
>
> ***Response***. Thanks for raising this important point. We would like to clarify that this is primarily part of the intended training formulation, rather than merely an implementation shortcut. In our model, the linear solver instantiates the prescribed coupled Bayesian update and therefore belongs to the definition of the flow-state generator $p_F(\cdot\mid z;t)$ (Sec. 4.1), while learning is performed only on the receiver predictor conditioned on the resulting belief states.
>
> In this sense, we keep the update geometry fixed and interpretable, and train the predictor under that geometry, rather than turning the solver itself into part of an end-to-end trainable state-construction pipeline. The computational benefit is secondary. Avoiding differentiation through a large SPD solve does reduce memory/time overhead, but the main reason is conceptual alignment with the formulation studied in this paper. We will clarify this point in the revision.

---

> > ### Author Rebuttal · Reviewer_7HL7 · 2026-04-04
> >
> > The authors have adequately addressed the main concerns raised in my original review. The rebuttal is clear, specific, and technically satisfactory, and it resolves the issues that were most important to my evaluation. Any remaining issues are minor and do not affect my overall positive assessment of the paper.

---

> > > ### Author Response · Authors · 2026-04-04
> > >
> > > Thank you very much for your careful reading and positive feedback. We truly appreciate your time and effort throughout the review process, and we are very glad that our response has addressed your main concerns.
> > >
> > > If you feel a score raise is appropriate, we would be very grateful.

---

### Official Review · Reviewer_y7sZ · 2026-03-13

**Soundness:** 3
**Presentation:** 3
**Significance:** 3
**Originality:** 3
**Overall Recommendation:** 4
**Confidence:** 4

**Summary:**

This paper presents a variational graph-generative model based on Bayesian Flow Networks (BFNs). It uses variational lifting, where each Bayesian update is abstracted as a symmetric, positive-definite linear system. The core idea for data processing is to use the graph to construct a dependency graph without label leakage and assign edge weights to form a weighted graph Laplacian. This is used to instantiate the prior precision and obtain the observation precision matrices. The training loss for a sender-receiver message-matching algorithm is defined in terms of the observation precision. The paper presents additional information about the SPD solvers used, the iteration complexity, and gradient computation and memory usage. The paper uses three graph generation models for synthetic experiments. The metrics used are V.U.N. and Ratio. Experiments on real data are performed on molecular graph generation (2 benchmarks). Both sets of experiments are performed using the proposed model and a non-trivial set of baselines.

**Compliance With Llm Reviewing Policy:**

Affirmed.

**Final Justification:**

The author's response is satisfactory, so I will maintain my positive score.

**Key Questions For Authors:**

I don't have many questions. Perhaps clarifications to the points above could be useful. I have just some additional clarification questions.

How much of the gain comes from the structured precision geometry versus simply adding another regularization mechanism to a strong BFN backbone?

Can the authors provide variance across random seeds and significance tests for the main benchmark improvements, especially where gains over DeFoG or TopBF are small?

**Limitations:**

The paper presents an Impact Statement arguing that improved graph-generation methods, such as VBFN, could accelerate applications such as molecular design, enabling faster drug discovery and promoting responsible use.
Not in the main body of the paper, but the authors included a Future Work section in the appendices that does provide a perspective about what could be done to improve the proposed VBFN model. I would suggest including a summarized version of this in the Conclusion

**Strengths And Weaknesses:**

The paper has a clear conceptual contribution: it identifies a real limitation of factorized BFNs for graphs and introduces a mathematically coherent alternative based on structured precisions and coupled Bayesian fusion. The transition from independent updates to an SPD linear system is clean and well-motivated. The paper states the main update theorem, shows the classical BFN case as a diagonal special case, gives an optimization interpretation of the posterior mean, and derives the KL simplification. The empirical section is broad enough to support the paper’s claims: it covers synthetic and molecular datasets, compares against strong autoregressive, diffusion, flow, and Bayesian-flow baselines, and includes meaningful ablations on coupling, observation precision, solver choice, efficiency, and step budget.
In terms of weaknesses, I would suggest more clarity in the exposition about what is optimized variationally and what is simply imposed architecturally. The variational framing seemed closer to a structured Gaussian design choice than to a deeply developed variational inference argument. Second, the empirical gains, while strong, are not always significantly greater against the best baselines. On some synthetic benchmarks, VBFN is close to DeFoG or TopBF rather than decisively better, so the evidence supports competitiveness more than absolute improvement. Finally, the practical scalability claim for larger graphs is weakened by memory usage. For instance, the sampling memory on ZINC250k is greater than GruM, although the time is comparable.

---

> ### Author Rebuttal · Authors · 2026-03-31
>
> # Response to Reviewer y7sZ
>
> We thank the reviewer for the constructive suggestions. We have carefully addressed all concerns below and will incorporate the corresponding clarifications, analyses, and revisions in the next version.
>
> Supplementary Experiments: https://anonymous.4open.science/r/VBFN-D6B0/Supplementary_Experiments.md
>
> ------
>
> > **Concern 1**: The interpretation of the term *variational*
>
> ***Response***. Thank you for raising this important point. We would like to clarify that our use of *variational* is specific rather than broad. We do not claim a general ELBO-based variational-inference framework. Instead, *variational* refers to **lifting of the admissible belief family** from the factorized Gaussian family in classical BFNs,
> $$
> q_t(z)=\mathcal N(z;\mu_t,\rho_t^{-1}I),
> $$
> to a tractable joint Gaussian family,
> $$
> q_t(z)=\mathcal N(z;\theta_t,P_t^{-1}),
> $$
> with structured precision and coupled posterior fusion. We will revise the text to make this narrower meaning explicit.
>
> ------
>
> > **Concern 2**: On whether VBFN is merely close to DeFoG/TopBF on synthetic benchmarks
>
> ***Response*.** Thanks for raising this concern. We agree that on some individual synthetic datasets the margins to strong recent baselines are not always large. However, our intended claim is about **cross-dataset consistency** rather than a single-dataset margin. According to Table 1, no previous method is top-2 on all three synthetic benchmarks, whereas VBFN is the only one consistently best or second-best across Planar, Tree, and SBM. On the molecular benchmarks, the advantage is stronger.
>
> ||Planar||Tree||SBM||
> | - | :-: | :-: | :-: | :-: | :-: | :-: |
> ||V.U.N.↑|Ratio↓|V.U.N.↑|Ratio↓|V.U.N.↑|Ratio↓|
> |DeFoG|**99.5**±1.0|1.6±0.4|96.5±2.6|1.6±0.4|**90.0**±5.1|4.9±1.3|
> |TopBF|98.3±1.7|1.8±0.4|95.2±3.3|1.7±0.8|89.2±4.9|3.3±1.5|
> |VBFN|99.5±0.9|**1.5**±0.6|**97.3**±2.4|**1.4**±0.5|89.7±2.3|**1.5**±0.7|
>
> We also followed your suggestion and repeated VBFN, TopBF, and DeFoG with 5 random seeds on the synthetic benchmarks where the margins are small. We now report mean ± std and two-sided Welch's t-tests. The ranking remains stable, supporting a claim of consistent competitiveness, with a statistically significant improvement on SBM/Ratio ($p<0.01$). The remaining gains are better interpreted as consistent rather than individually large-margin. We will revise the wording accordingly.
>
> ------
>
> > **Concern 3**: On how much of the gain comes from structured precision geometry versus adding another regularization mechanism
>
> ***Response***. Thanks for this important question. We hope the inclusion of TopBF helps provide a useful perspective. As a critical baseline, TopBF is a closely related BFN-family baseline with a highly consistent experimental setup, including **core network, CDF-based parameterization, and other overlapping parameters**. TopBF introduces graph structure through an external regularization route, whereas VBFN makes structure intrinsic to Bayesian belief evolution itself. Concretely, VBFN updates the belief state through
> $$
> P_t \triangleq \Omega_{\mathrm{prior}}+\beta(t)\Omega_{\mathrm{obs}}, \qquad
> \theta_t=P_t^{-1}\left(\Omega_{\mathrm{prior}}\theta_0+\beta(t)\Omega_{\mathrm{obs}}y\right).
> $$
> Thus, the graph-aware geometry enters the posterior update directly. Empirically, VBFN performs better overall than TopBF, especially on the molecular benchmarks. This interpretation is further supported by our coupling ablations: weakening or removing the coupling degrades performance, suggesting that the gain comes not merely from adding another regularization term to a strong BFN backbone, but from making coupling intrinsic to the Bayesian update itself.
>
> ---
>
> > **Concern 4**: On scalability and memory
>
> ***Response***. We agree that practical scalability should consider both runtime and memory. On ZINC250k, VBFN indeed uses more sampling memory than GruM, so we will revise the paper to avoid overstating efficiency. Our intended claim is that VBFN remains feasible on larger molecular graphs with competitive wall-clock time, but at a clear memory cost due to the structured, non-factorized belief update.
>
> To make this more explicit, we added two supplementary analyses. A stage-wise memory profile shows that the dominant persistent cost comes from the dense edge-space precisions and posterior cache, whereas the largest transient peaks arise in Bayes fusion and predictor/decoder stages. An additional comparison with GruM, DeFoG, and TopBF shows that VBFN(CG) stays competitive in training/sampling time on ZINC250k, while its sampling memory is noticeably higher. Detailed results are provided in the anonymous supplementary material (Supplementary Tables 3 and 4).
>
> We will revise the paper to present this more clearly as a performance-memory trade-off, discuss that the current implementation is not yet memory-optimized, and summarize these limitations and future optimization directions in the Conclusion.

---

> > ### Author Rebuttal · Reviewer_y7sZ · 2026-04-04
> >
> > Thanks for your reply. This clarifies my questions.

---

> > > ### Author Response · Authors · 2026-04-04
> > >
> > > Thank you very much for your careful reading and positive feedback. We truly appreciate your time and effort throughout the review process, and we are glad that our response has clarified your concerns.
> > >
> > > If you feel a score update is appropriate, we would be very grateful.

---

### Decision · Program_Chairs · 2026-04-30

**Decision:**

Accept (regular)

**Comment:**

The reviewers found the method technically solid, competitive on the synthetic benchmarks, and strong on the molecular datasets, with ablations suggesting that the node-edge coupling is genuinely useful. The main concerns were the narrower-than-initially-stated scope, the high memory cost on larger graphs, and the hand-designed coupling structure. The rebuttal helped by narrowing the claims, improving the empirical evaluation pipeline (e.g. testing multiple different random seeds), and promising to be more direct about the runtime-memory tradeoff and current limitations in the potential camera-ready. In the end, all reviewers vote for acceptance and I support.